# Abnormal mGluR-mediated synaptic plasticity and autism-like behaviours in *Gprasp2* mutant mice

Mohamed Edfawy [1,2,3], Joana R. Guedes [1,2], Marta I. Pereira[1], Mariana Laranjo[1], Mário J. Carvalho[1], Xian Gao[4,5,6], Pedro A. Ferreira [1], Gladys Caldeira[1,2], Lara O. Franco[1,2,3], Dongqing Wang[4], Ana Luisa Cardoso [1,2], Guoping Feng [4,5,6], Ana Luisa Carvalho [1,7] & João Peça [1,2]

Autism spectrum disorder (ASD) is characterized by dysfunction in social interactions, stereotypical behaviours and high co-morbidity with intellectual disability. A variety of syndromic and non-syndromic neurodevelopmental disorders have been connected to alterations in metabotropic glutamate receptor (mGluR) signalling. These receptors contribute to synaptic plasticity, spine maturation and circuit development. Here, we investigate the physiological role of *Gprasp2*, a gene linked to neurodevelopmental disabilities and involved in the postendocytic sorting of G-protein-coupled receptors. We show that *Gprasp2* deletion leads to ASD-like behaviour in mice and alterations in synaptic communication. Manipulating the levels of *Gprasp2* bidirectionally modulates the surface availability of mGluR$_5$ and produces alterations in dendritic complexity, spine density and synaptic maturation. Loss of *Gprasp2* leads to enhanced hippocampal long-term depression, consistent with facilitated mGluR-dependent activation. These findings demonstrate a role for *Gprasp2* in glutamatergic synapses and suggest a possible mechanism by which this gene is linked to neurodevelopmental diseases.

[1] CNC - Center for Neuroscience and Cell Biology, University of Coimbra, 3004-504 Coimbra, Portugal. [2] Institute for Interdisciplinary Research (IIIUC), University of Coimbra, 3030-789 Coimbra, Portugal. [3] PhD Program in Experimental Biology and Biomedicine (PDBEB), University of Coimbra, 3030-789 Coimbra, Portugal. [4] McGovern Institute for Brain Research, Department of Brain and Cognitive Sciences, Massachusetts Institute of Technology, Cambridge, MA 02139, USA. [5] Key Laboratory of Brain Functional Genomics, Institute of Cognitive Neuroscience, School of Psychology and Cognitive Science, East China Normal University, Shanghai 200062, China. [6] Stanley Center for Psychiatric Research, Broad Institute of MIT and Harvard, Cambridge, MA 02142, USA. [7] Department of Life Sciences, University of Coimbra, 3004-517 Coimbra, Portugal. These authors contributed equally: Joana R. Guedes, Marta I. Pereira. Correspondence and requests for materials should be addressed to J.Pça. (email: jpeca@cnc.uc.pt)

While genetic studies of ASD point to a complex and heterogeneous aetiology[1–3], common signalling pathways linked to this disorder include elements important for synapse formation and the regulation of synaptic transmission[4–8]. One of the most salient findings in animal models of ASD and intellectual disability (ID) is the presence of abnormal mGluR5-mediated synaptic plasticity[9]. These alterations are observed in animal models of Fragile X (FMRP)[10], Phelan-McDermid (SHANK3)[11,12], Tuberous Sclerosis (TSC1/TSC2)[13,14], Cowden (PTEN)[15], 16p11.2 microdeletion[16] and Rett syndrome (MECP2)[17]. Type I mGluRs play an important role in synaptic plasticity in the cerebellum[18,19] and hippocampus[20,21] where their activation leads to internalization of α-amino-3-hydroxy-5-methyl-4-isoxazolepropionic acid (AMPA) receptors. In particular, mGluRs are essential in the removal and weakening of spines[22], circuit remodelling in the mouse somatosensory cortex[23] and in experience-dependent synaptic maturation[24].

At the synapse, mGluRs are anchored by SHANK and HOMER proteins in a complex that clusters together ionotropic and metabotropic glutamate receptors[25]. Deletion of *Shank2* or *Shank3* leads to autism-like behaviour and perturbations in synaptic comunication[26,27], while altered mGluR5-HOMER scaffolds play a major contribution in the phenotype and synaptic deficits in *Fmr1* knockout (KO) mice[28]. At the same time, like most G-protein-coupled receptors (GPCRs), mGluRs are highly regulated and desensitize following agonist binding[29]. However, the role played by the endocytic sorting partners that determine the recycling or the degradation of these receptors remains underexplored[30,31].

The G-protein-coupled receptor associated sorting proteins (GPRASPs) comprise a large family of genes that interact and regulate trafficking of GPCRs[32] such as delta opioid receptor[33], D2 dopamine receptor[34], muscarinic receptors, and mGluR1 and mGluR5 receptors[35]. GPRASP1-10 have been proposed to act in a cell type-specific manner to control the postendocytic fate of GPCRs[30]. From this large family of genes, *GPRASP2* is located within the Xq22.1 deletion syndrome region associated with severe ID in humans[36,37]. In addition, *GPRASP2* mutations have been found in autism and schizophrenia patients[38,39] and the downregulation of this gene was observed in a large cohort of brain samples from autism patients[40]. We hypothesized that *Gprasp2* deletion could perturb key elements in synaptic maturation and induce alterations in synaptic activity in brain circuits relevant to ASD and ID by perturbing GPCR physiology. Using a combination of biochemical, electrophysiology, imaging and behavioural analysis, we show that GPRASP2 interacts and regulates mGluR5, that deletion of *Gprasp2* alters synaptic comunication and enhances mGluR-long-term depression (LTD) in hippocampal circuits, and that *Gprasp2* knockout (KO) mice exhibit autism-like behaviours.

## Results

### *Gprasp2* knockout alters hippocampal neuronal morphology.
In order to understand the role of GPRASP2 in vivo we generated a novel conditional mouse line. For this, a germline deletion was created by crossing conditional *Gprasp2* F1 mice with a beta-Actin-Cre driver line (Fig. 1a). Confirmation of the null allele was performed using western blot, PCR, mRNA in situ hybridization and qRT-PCR (Fig. 1a–c and Supplementary Fig. 1a–b). We were able to amplify *Gprasp* family members from the brain of P20 animals (with the exception of *Gprasp5*) and did not observe changes in their expression levels when comparing KO and littermate controls (Supplementary Fig. 1b). *Gprasp2* KO mice backcrossed for at least five generations into C57/BL6 background were viable and did not display gross abnormalities with the

exception of a significant increase in body weight starting from 12 weeks of age (Supplementary Fig. 1b–c). In this work, we studied male *Gprasp2−/y* and *Gprasp2+/y* littermates during the juvenile period (6- to 10-week-old), before significant physiological differences in body weight manifested (Supplementary Fig. 1c). In wild-type (WT) mice, GPRASP2 protein levels peaked during postnatal days P15–P20 (Fig. 1d), which is an active period for synaptic contact refinement[41]. In terms of regional expression, *Gprasp2* mRNA was enriched in the hypothalamus and hippocampus of young and adult mice (Fig. 1e–g and Supplementary Fig. 2a–i) and transiently high levels in the striatum of newborn (P5) mice (Supplementary Fig. 2j–k). Regarding protein levels, GPRASP2 was found to be highly expressed in the hippocampus and hypothalamus (Supplementary Fig. 2l). In humans, GPRASPs show wide tissue distribution (see Supplementary Table 1), but GPRASP1-3 are the family members most expressed in the human brain.

Since *Gprasp2* was enriched in the mouse hippocampus, we started by exploring if there were alterations in this region in the KO animals. We investigated the morphology of CA1 pyramidal neurons using an in vivo Golgi-like labelling, via peripheral injection of AAV9.hSyn.GFP (see Methods section). This procedure allowed for an unbiased labelling of neurons with well-separated dendritic branches (Fig. 1h). Using Sholl analysis of reconstructed CA1 pyramidal neurons we found a reduction in dendritic arborisation and a decrease in total dendritic length in distal regions of mutant pyramidal neurons (Fig. 1i). Analysis of basal and apical dendritic complexity also revealed a more striking effect on apical neuronal arborization (Supplementary Fig. 3a–b). Taking into account the expression of *Gprasp2* and the increase weight gain displayed by the KO mice, we analysed the hypothalamus for changes in neuronal complexity and dendritic spine density but found no alterations in either parameter (Supplementary Fig. 4a–c).

### Functional alteration in synaptic transmission in *Gprasp2−/y* mice.
Next, we investigated the hippocampus for functional alterations emerging from the deletion of *Gprasp2* and recorded field excitatory postsynaptic potentials (fEPSPs) in the CA1 area while stimulating Schaffer collateral fibres (Fig. 2a). We found that *Gprasp2−/y* mice presented significantly reduced fEPSP amplitudes when compared with WT littermates (Fig. 2a, WT: $0.938 \pm 0.104$ mV, $n = 7/3$ slice/mice; KO: $0.615 \pm 0.062$ mV, $n = 5/3$ slice/mice; at $140\,\mu A$; values expressed as mean ± s.e.m.). To dissect this alteration and determine if presynaptic function was compromised, we measured the amplitude of negative peak 1 response (NP1; a measure of fibre depolarization) and assessed paired-pulse ratios (Fig. 2b, c). No overt alterations in either parameter was identified, suggesting that field response deficits could arise from postsynaptic impairment in the CA3-CA1 circuit of *Gprasp2−/y* mice.

To further characterize the alterations in glutamatergic transmission caused by *Gprasp2* deletion we performed whole-cell patch-clamp recordings in CA1 pyramidal neurons in P15–P20 *Gprasp2−/y* and *Gprasp2+/y* littermates. We show that the amplitude of AMPA receptor-mediated miniature excitatory postsynaptic currents (mEPSCs) was significantly reduced in KO mice (Fig. 2d–f) (WT: $13.03 \pm 0.307$ pA, $n = 23/4$ cells/mice; KO: $11.20 \pm 0.424$ pA, $n = 25/4$ cells/mice; mean ± s.e.m.). These results suggest a postsynaptic impairment in CA1 synapses. However, we did not observe significant alterations in frequency of mEPSCs (Fig. 2g) or in the kinetics of the miniature events (Fig. 2h, i). In line with the previous results, we found a reduction in the scaffolding protein PSD-95 and in AMPA receptor subunits GluA1 and GluA2 in synaptosomal plasma membrane fraction from KO mice (Fig. 2j).

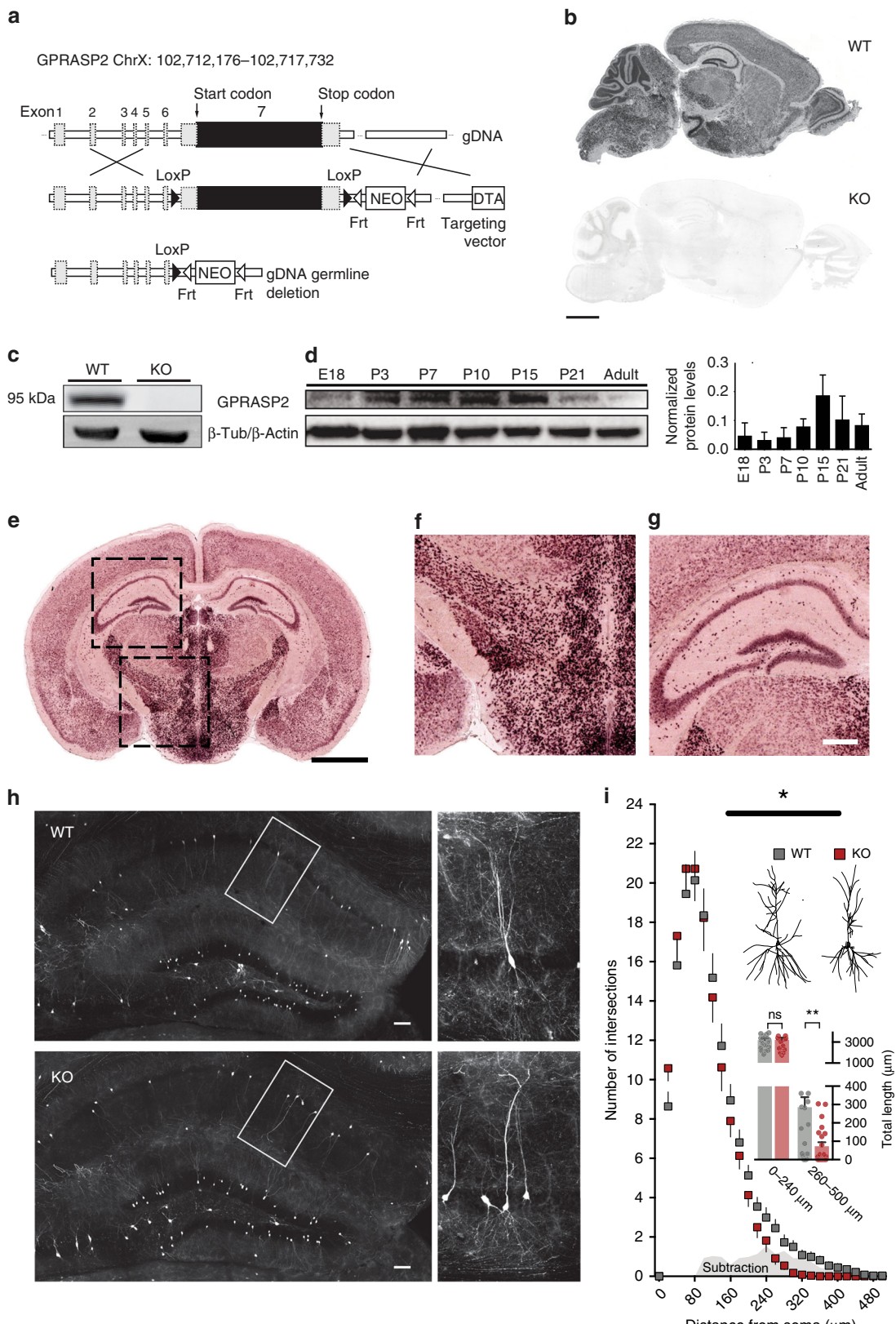

**Gprasp2 deletion impairs neuronal spine maturation**. To understand if *Gprasp2* is required for neuronal spine maturation in vivo, we analysed spine density in GFP-labelled CA1 pyramidal neurons (Fig. 3a). While no significant alterations in overall spine density counts in apical dendrites were identified (Fig. 3b), a breakdown of spine categories in mature and immature types

revealed a decrease in density of mature spines in apical regions of *Gprasp2*$^{-/y}$ mice (Fig. 3b). Spine density in basal dendrites alone did not show statistically significant differences; however, increasing statistical power by combining results from apical and basal dendritic segments revealed a decrease in total spine density in *Gprasp2* KO mice (Supplementary Fig. 3c–d). When

**Fig. 1** *Gprasp2* knockout mice display structural alterations in hippocampal neurons. **a** *Gprasp2* deletion targeting strategy. **b** In situ hybridization for *Gprasp2* mRNA detects strong signal in P15 WT mice and no signal in *Gprasp2^{−/y}* mice. Scale bar, 1 mm. **c** Western blotting shows the deletion of GPRASP2 in whole-brain lysates from KO mice. **d** Expression pattern of GPRASP2 in brain lysates of WT mice from E18 to 3-month of age; $n = 3$ mice. **e–g** Coronal section from P15 WT mice reveals *Gprasp2* is highly expressed in (**f**) the hypothalamus and in (**g**) the hippocampal formation. Scale bar, 2 mm in (**e**) and 250 μm in (**f–g**). **h** Representative images from the CA1 region in AAV9.hSyn.GFP infected mice. High-magnification images of *Gprasp2^{−/y}* and WT CA1 pyramidal neurons. Scale bar, 100 μm. **i** Sholl analysis reveals decreased neuronal complexity of *Gprasp2^{−/y}* CA1 pyramidal neurons compared with WT littermates; WT $n = 22/3$ neurons/mice, KO $n = 22/3$ neurons/mice; two-way repeated measures ANOVA. Inset (top), representative CA1 neurons. Inset (bottom), total dendritic length in distal (260–500 μm) but not proximal (0–240 μm) regions is reduced in *Gprasp2^{−/y}*; one-way ANOVA with Bonferroni post hoc. All data are presented as mean ± s.e.m. Statistical significance: *$p < 0.05$, **$p < 0.01$

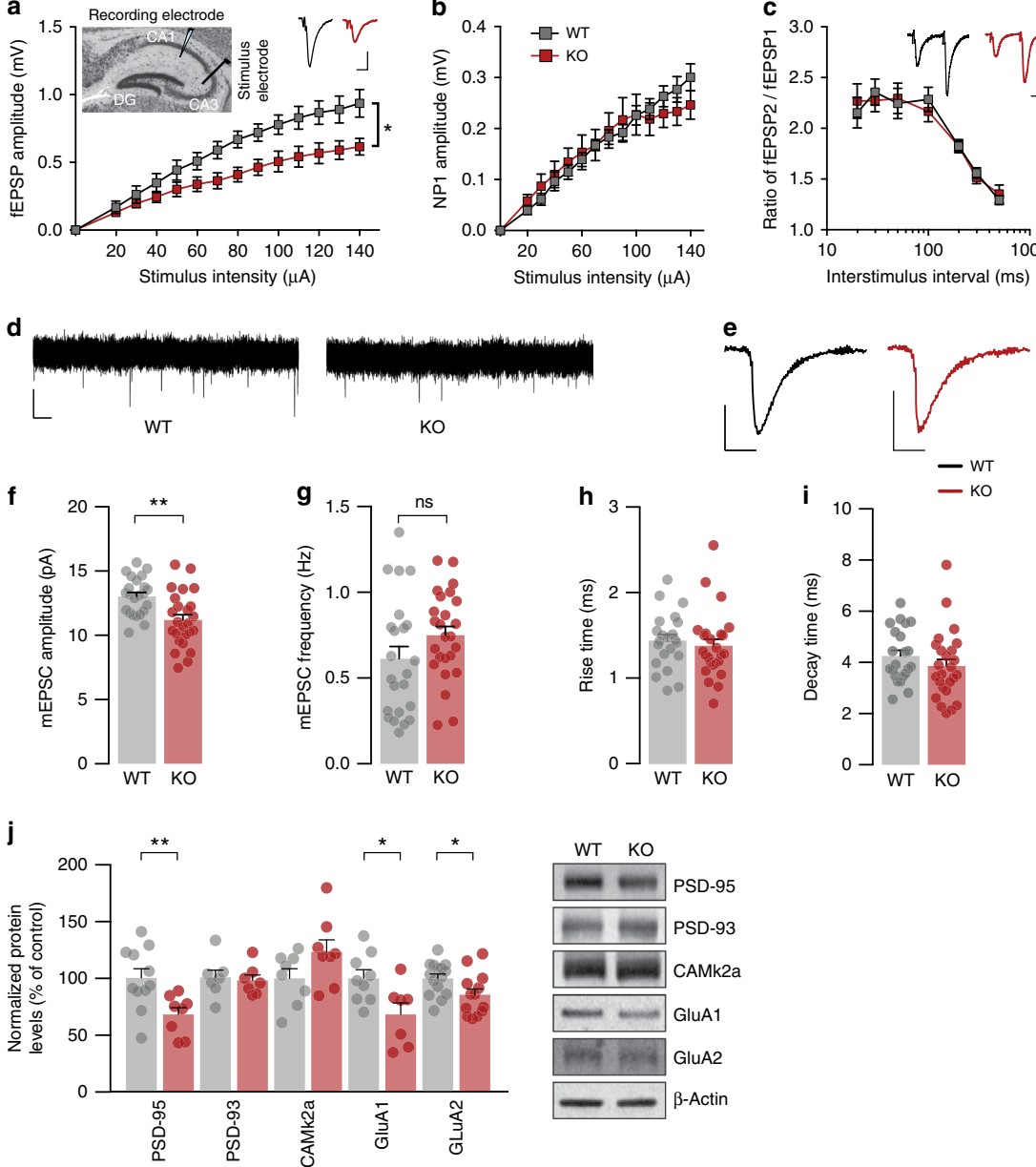

**Fig. 2** Deletion of *Gprasp2* impairs glutamatergic transmission. **a** Input–output curve shows fEPSPs are reduced in the hippocampus of *Gprasp2^{−/y}*; WT $n = 7/3$ slice/mice, KO $n = 5/3$ slices/mice; two-way repeated measures ANOVA. Inset, representative traces. Scale bar, 0.5 mV, 10 ms. **b** NP1 amplitude is unaltered between KO and WT littermates; WT $n = 7/3$ slices/mice, KO $n = 5/3$ slices/mice. **c** Paired-pulse facilitation is not altered in *Gprasp2* KO mice; WT $n = 11/7$ slices/mice, KO $n = 7/7$ slices/mice. Inset shows representative traces. Scale bar, 0.5 mV, 10 ms. **d** mEPSC traces from *Gprasp2^{−/y}* and WT littermates. Scale bar, 10 pA, 1 s. **e** Representative mEPSC average traces. Scale bar, 5 pA, 10 ms. **f, g** Reduced mEPSC amplitude (**f**) but not frequency (**g**) in *Gprasp2^{−/y}* CA1 pyramidal neurons; WT $n = 23/4$ cells/mice, KO $n = 25/4$ cells/mice; two-tailed unpaired *t*-test. **h, i** Rise (**h**) and decay (**i**) times for mEPSCs is unchanged in *Gprasp2^{−/y}* and WT mice; WT $n = 22/4$, KO $n = 25/4$; two-tailed Mann–Whitney test. **j** Reduced levels of AMPA receptors and PSD-95 in hippocampal synaptosomal plasma membrane fraction in *Gprasp2^{−/y}* mice; WT $n = 7$–15, KO = 7–13; two-tailed unpaired *t*-test. Data are presented as means ± s.e.m. Statistical significance: *$p < 0.05$, **$p < 0.01$

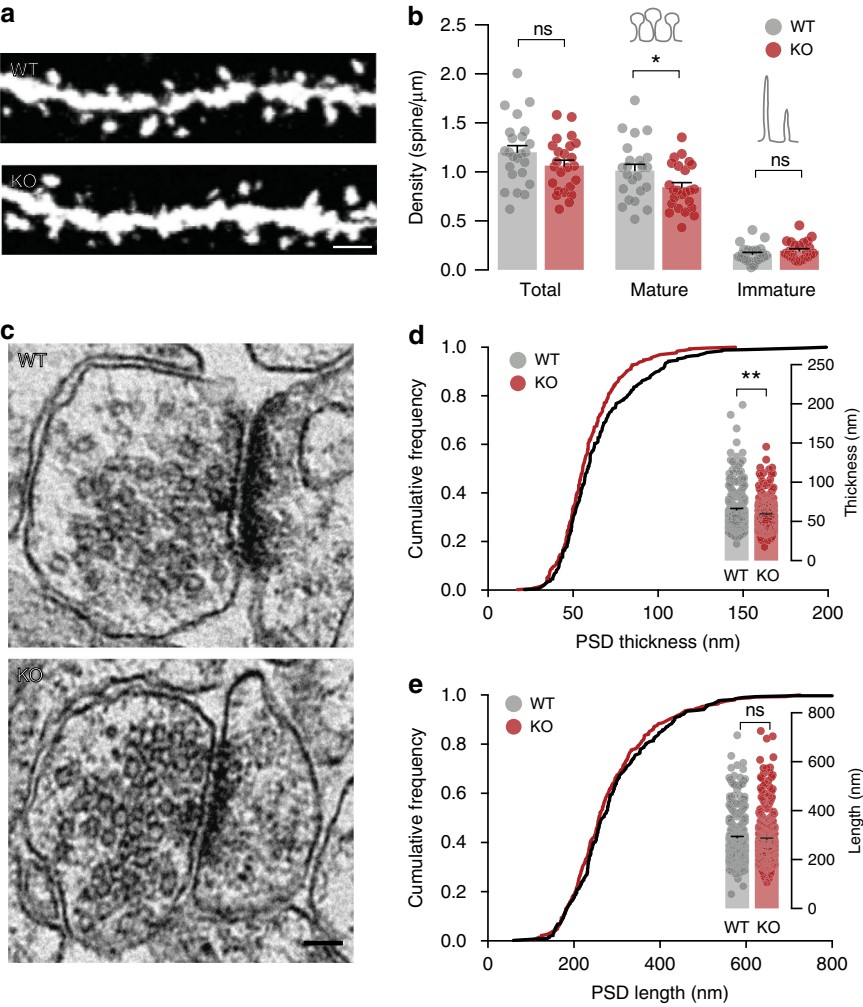

**Fig. 3** Impaired spine maturation in hippocampal spines in *Gprasp2* KO mice. **a** Representative images from secondary dendrites of CA1 hippocampal neurons from *Gprasp2*$^{-/y}$ mice and WT controls. Scale bar 1 μm. **b** Reduced density of mature spines in secondary apical branches of hippocampal CA1 neurons from *Gprasp2*$^{-/y}$ mice; WT $n = 24/3$ branches/mice, KO $n = 24/3$ branches/mice; two-tailed *t*-test. **c** Representative electron micrographs from hippocampal PSDs from *Gprasp2*$^{-/y}$ and WT mice. Scale bar, 100 nm. **d**, **e** *Gprasp2*$^{-/y}$ mice present thinner PSDs (**d**) but no significant alterations in PSD length (**e**) when compared with WT animals; WT $n = 337/3$ PSDs/mice; KO $n = 362/3$ PSDs/mice, two-tailed Mann–Whitney test. Data in (**d**) and (**e**) (main graph) are presented as a cumulative frequency plot, all other data are presented as means ± s.e.m. Statistical significance: *$p < 0.05$, **$p < 0.01$

comparing results from basal and apical dendrites, this could suggest that spine density decrease in KO neurons is more pronounced in distal segments.

To validate the alterations in spine structure, we used electron microscopy to perform an ultrastructural analysis of postsynaptic densities (PSD) from the CA1 region. Our experiments revealed a decrease in the thickness of PSDs from *Gprasp2*$^{-/y}$ mice (Fig. 3c–e), again pointing to alterations in postsynaptic composition.

Interestingly, changes in spine maturation are strongly implicated in several neurodevelopmental disorders[42] and stable spines express greater numbers of AMPA-type receptors[43]. Therefore, our functional and structural analysis indicate that loss of *Gprasp2* weakens synaptic contacts and perturbs spine maturation.

**GPRASP2 levels bidirectionally impact neuronal arborization and spine density**. Since compensation effects during brain development may occlude more striking synaptic alterations, we turned to dissociated primary cell cultures to further dissect the

role of GPRASP2 in neuronal morphology and spine development. This way, we could perform acute manipulation to GPRASP2 and determine whether the alterations observed arise from a combination of cell autonomous and cell non-autonomous phenomena. We started by performing GPRASP2 knockdown studies using shRNA to determine the consequences of its decreased expression of in developing rat hippocampal neurons (Fig. 4a). We found that neuronal complexity and total dendritic length were significantly reduced in neurons expressing a shRNA targeting the endogenous rat *Gprasp2*. These alterations could be rescued by co-expression of the murine form of GPRASP2 (which is not affected by the shRNA) (Fig. 4b, c). Acute knockdown of GPRASP2 very significantly reduced spine density (Fig. 4d, e), but did not induce changes to overall spine shape (Fig. 4f–h). This experiment led to a modest significant increase in overall spine length and head diameter, suggesting a potential overshoot in terms of expression levels that could influence these spine parameters. To exclude other potential off-target effects of our knockdown system we also tested a second shRNA (shRNA II) targeting a different region of the GPRASP2 mRNA (Supplementary Fig. 5). Despite being less effective in

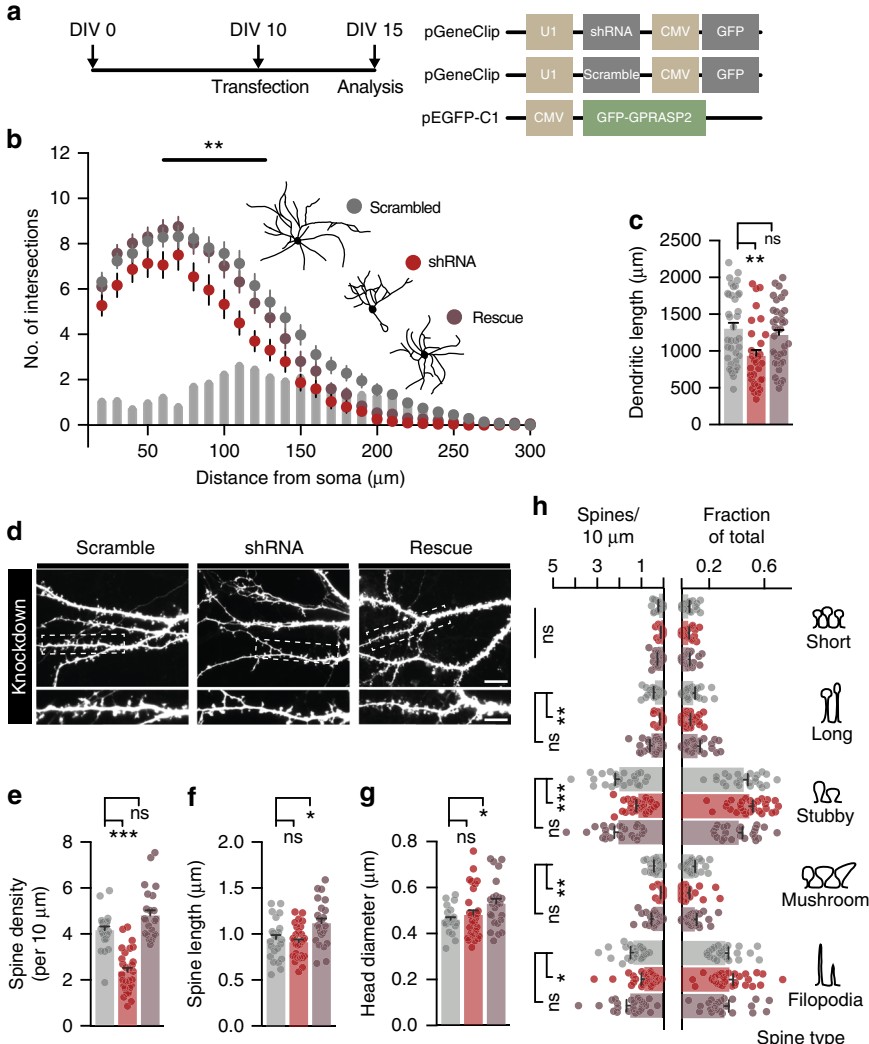

**Fig. 4** Cell-autonomous reduction in dendritic complexity and spine density in acute GPRASP2 knockdown. **a** Cultured hippocampal neurons were transfected with scramble shRNA (Scramble), shRNA against *Gprasp2* (shRNA) or GFP-GPRASP2 + shRNA (rescue) at DIV10 and analysed at DIV15. **b** Sholl analysis reveals reduced dendritic complexity in shRNA transfected neurons when compared with Scramble (effect of treatment **$p < 0.01$) and rescue (effect of treatment *$p < 0.01$); scramble $n = 38$, shRNA $n = 30$, rescue $n = 37$, from three independent preparations; two-way repeated measures ANOVA with Bonferroni post hoc test. **c** GPRASP2 knockdown decreases total dendritic length when compared with scramble and rescue transfected neurons; scramble $n = 38$, shRNA $n = 30$, rescue $n = 37$, from three independent preparations; one-way ANOVA test followed by Tukey multiple comparisons test. **d–g** GPRASP2 knockdown leads to decreased spine density (**e**), but no change in spine length (**f**) or head diameter (**g**) when comparing shRNA to scramble condition; scramble $n = 24$, shRNA $n = 29$, rescue $n = 24$, from three independent preparations; one-way ANOVA test followed by Tukey multiple comparisons test. **h** When divided into spine types, dendritic spine density was decreased across most spine types following GPRASP2 overexpression, but no changes are observed when normalized to fractional counts; scramble $n = 24$, shRNA $n = 29$, rescue $n = 24$, from three independent preparations; one-way ANOVA test followed by Tukey multiple comparisons test. Data are presented as means ± s.e.m. Statistical significance: *$p < 0.05$, **$p < 0.01$ and ***$p < 0.001$

reducing GPRASP2 expression (Supplementary Fig. 5a–c), shRNA II led to reduced neuronal complexity and spine density in rat hippocampal neuron cultures. These results highlight a considerable influence of GPRASP2 expression in neuronal morphology and spine maturation.

Next, we assessed if overexpression of GPRASP2 could also modulate spine and neuronal morphology. Interestingly, we found that increasing the expression levels of GPRASP2 promoted enhanced neuronal dendritic arborisation and increased dendritic length (Fig. 5a–c). Moreover, spine density, spine length and spine head diameter were significantly increased in GPRASP2-overexpressing neurons (Fig. 5d–g), consistent with a shift of spine morphology towards more mature spine types and reduction of filopodia-like spines (Fig. 5h). Together, these

observations suggest that *Gprasp2* expression levels bidirectionally alter neuronal spine density and spine maturation in a cell-autonomous manner, contributing to neuronal development and influencing neuronal dendrite and spine maturation.

**GPRASP2 modulates mGluR$_5$ surface availability**. Since GPRASPs have been proposed to regulate mGluRs[35], and since these receptors play a critical role in plasticity, spine morphology, synaptic communication[18–22] and neurite elongation[44], we decided to investigate if alterations in mGluR$_5$ could explain some of the changes we observed when manipulating *Gprasp2*. Considering the known role of GPRASPs in postendocytic sorting, we assessed if GPRASP2 and mGluR$_5$ interact and also, if the co-

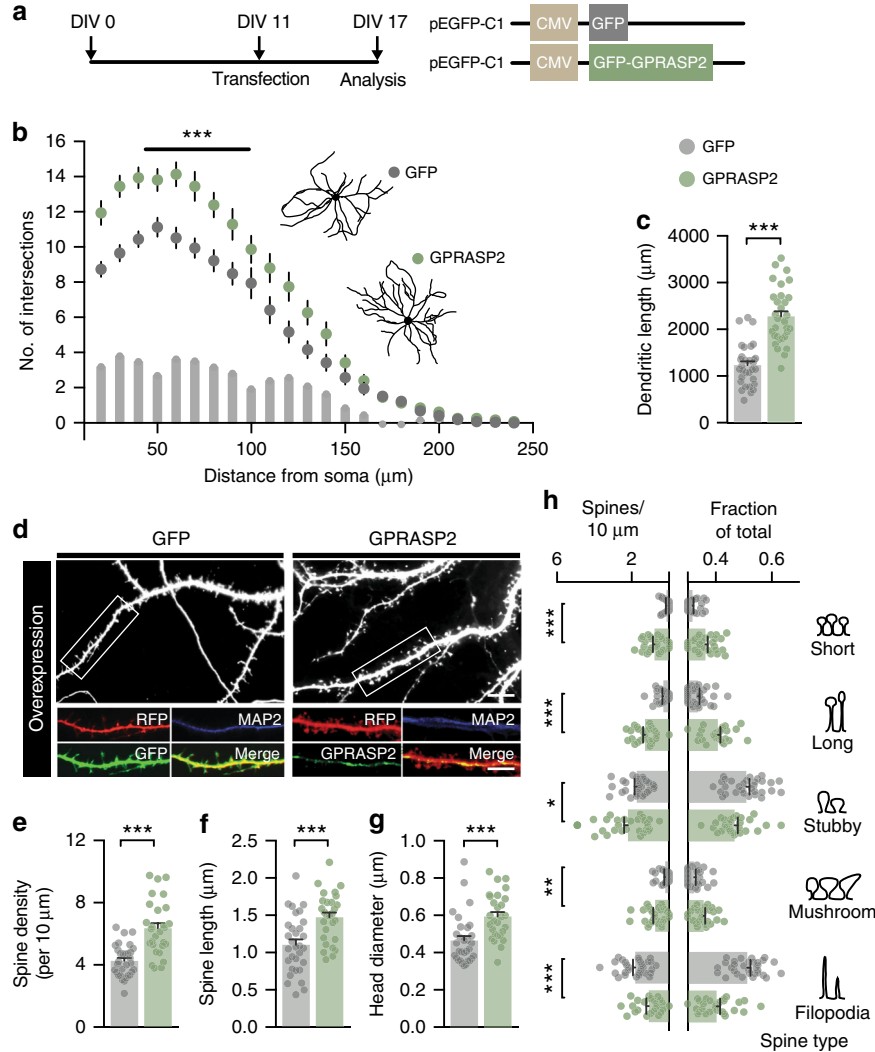

**Fig. 5** GPRASP2 overexpression induces cell autonomous increased in spine density and spine maturation. **a** Cultured hippocampal neurons were transfected with GFP alone or GFP-GPRASP2 at DIV11 and analysed at DIV17. **b** Sholl analysis reveals increased dendritic complexity in GPRASP2-overexpressing neurons compared with GFP controls; GFP $n = 31$, GPRASP2 $n = 31$, from three independent preparations; two-way repeated measures ANOVA. **c** GPRASP2 overexpression increases total dendritic length; GFP $n = 31$, GPRASP2 $n = 31$, from three independent preparations; two-tailed $t$-test. **d–g** GPRASP2 overexpression leads to increased spine density (**e**), increased spine length diameter (**f**) and increased spine head diameter (**f**); GFP $n = 31$, GPRASP-GFP $n = 28$ from three independent preparations; two-tailed $t$-test. Scale bars 2 μm. **h** Dendritic spine density was increased following GPRASP2 overexpression, with the exception of filopodia-like spines, which showed a significant decrease; GFP $n = 31$, GPRASP-GFP $n = 28$ from three independent preparations; one-way ANOVA test followed by Tukey multiple comparisons test. Data are presented as means ± s.e.m. Statistical significance: $^*p < 0.05$, $^{**}p < 0.01$ and $^{***}p < 0.001$

localization of GPRASP2 with markers of endocytic pathways is influenced by receptor activation. Using co-immunoprecipitation, we were able to pull-down mGluR$_5$ together with GPRASP2 in transfected HT-22 cells (a mouse hippocampal neuronal cell line) (Supplementary Fig. 6a). This biochemical interaction was not perturbed with receptor activation following 5- or 30-min of stimulation with 3,5-Dihydroxyphenylglycine (DHPG, an mGluR$_5$ receptor agonist). We then tested if stimulation with DHPG would alter the co-localization of GPRASP2 with other endocytic proteins such as Clathrin light chain, Rab5, Rab7 and Lamp1. We found that a 5-min stimulation significantly increased the overlap between GPRASP2 and Lamp1, which is consistent with a role for GPRASP2 in lysosomal trafficking after agonist-mediated receptor internalization[30,33] (Supplementary Fig. 6b).

Next, to functionally test if the *Gprasp2* knockdown-induced alterations in neuronal morphology could be due to mGluR$_5$

activity, we repeated our in vitro experiments in the presence of MPEP (a selective antagonist for mGluR$_5$) from DIV10 to DIV14. We found that MPEP treatment prevented *Gprasp2* knockdown-mediated reduction in neuronal arborization and recovered spine density to control conditions (Supplementary Fig. 7a–f). Together, these results point to a mechanism whereby GPRASP2 leads to changes in mGluR-dependent activity by perturbing receptor trafficking.

To better understand the consequences of bidirectionally altering the levels of GPRASP2 on mGluR physiology, we performed acute manipulations in vitro in the presence or absence of DHPG stimulation (100 μM for 30 min) and assessed changes to the endogenous levels of surface mGluR$_5$ and total mGluR$_{1/5}$ (Fig. 6a–f). Our results showed that acute overexpression of GPRASP2 was sufficient to decrease surface mGluR$_5$ signal by 23 ± 4.57% compared with GFP expressing-control neurons (Fig. 6c).

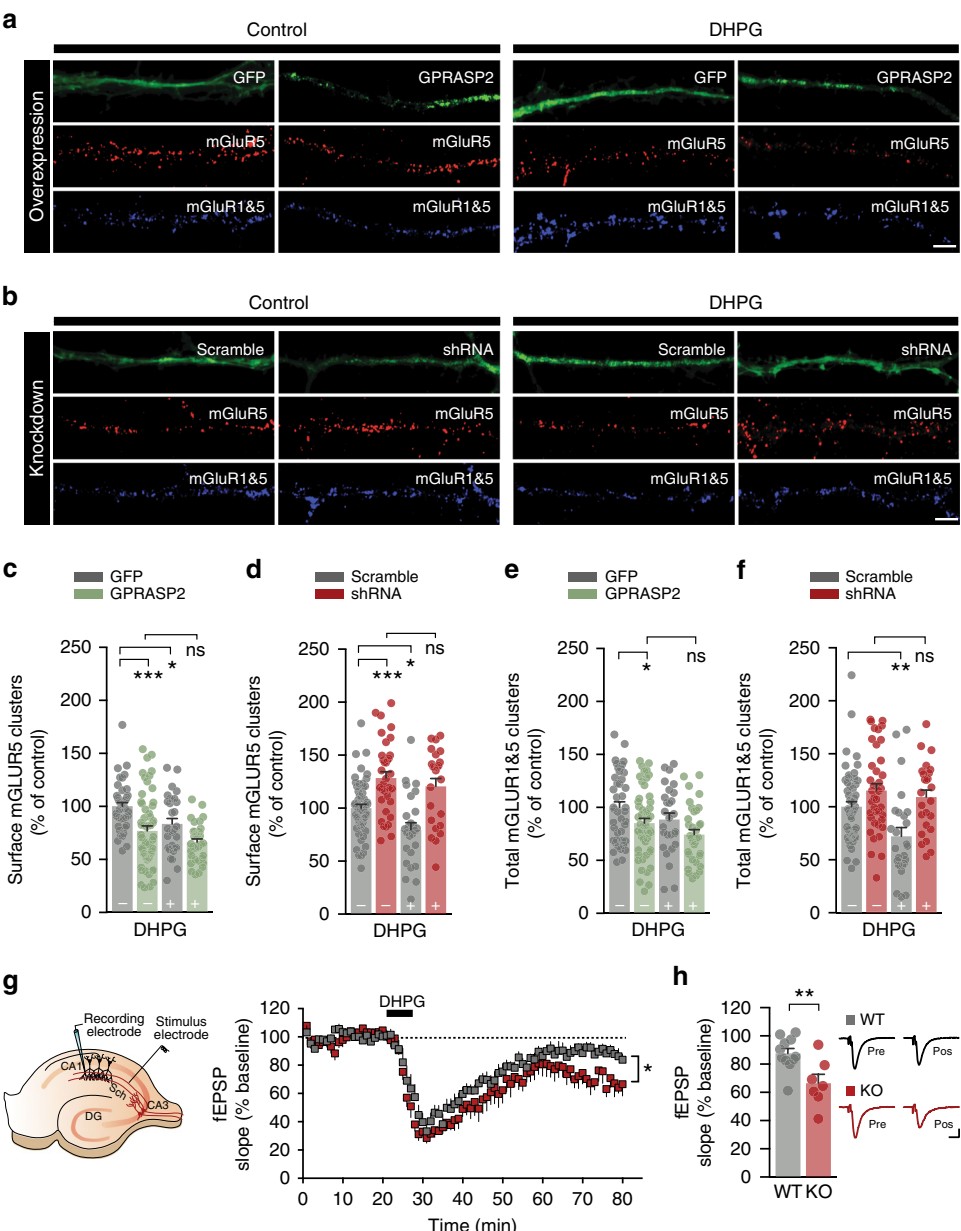

**Fig. 6** Bidirectional regulation of mGluR$_5$ surface availability and enhanced mGluR-LTD in *Gprasp2$^{-/y}$* mice. **a, b** Representative images from hippocampal neurons transfected with either GFP/GFP-GPRASP2 (**a**) or scramble/shRNA (**b**) in presence or absence of DHPG. Scale bar, 2 μm. **c, d** Reduced surface mGluR$_5$ clusters are observed following overexpression of GPRASP2 (**c**), while there is an increased in surface mGluR$_5$ clusters in GPRASP2 knockdown conditions (**d**), compared with control cells, with or without the effect of DHPG. **e, f** Reduced total mGluR$_{1/5}$ clusters under overexpression of GPRASP2 (**e**) and increased total mGluR$_{1/5}$ clusters in GPRASP2 knockdown conditions (**f**) when compared with control cells, with or without the effect of DHPG. Results are expressed as % of control cells for GPRASP2 overexpression vs GFP (control) or shRNA vs scramble (control); $n$ = 21–45 per condition from four independent preparations; one-way ANOVA test with Sidak´s post hoc test. **g** Schematic diagram of position of the stimulating and recording electrodes in the CA3-CA1 Schaffer collaterals circuit. When compared with WT littermates, *Gprasp2$^{-/y}$* mice display enhanced mGluR-dependent LTD upon DHPG stimulation; WT $n$ = 11/7 slices/mice, KO $n$ = 7/7 slices/mice; two-way repeated measures ANOVA. **h** fEPSP slope in the last 5 min of the recording post-DHPG stimulation; WT $n$ = 11/7 slices/mice, KO $n$ = 7/7 slices/mice; two-tailed *t*-test. Scale bar, 0.2 mV, 20 ms. All data are presented as mean ± s.e.m. Statistical significance: *$p$ < 0.05, **$p$ < 0.01 and ***$p$ < 0.001

Moreover, we found that upon overexpression of GPRASP2, the agonist-induced reduction in mGluR$_5$ surface expression could be partially occluded (Fig. 6c; $p$ = 0.053) (Fig. 6c). Conversely, the knockdown of GPRASP2 increased surface mGluR$_5$ clusters by 28 ± 3.0% and effectively blocked DHPG-mediated changes in receptor levels (Fig. 3d, f). This suggests that loss of GPRASP2

perturbs the normal mechanism of receptor internalization upon agonist-mediated activation.

Finally, to determine if the loss of *Gprasp2* alters mGluR$_5$-dependent plasticity in vivo, we probed *Gprasp2$^{-/y}$* mice and littermate controls for DHPG-mediated LTD in acute hippocampal slices. In these experiments, we found that DHPG

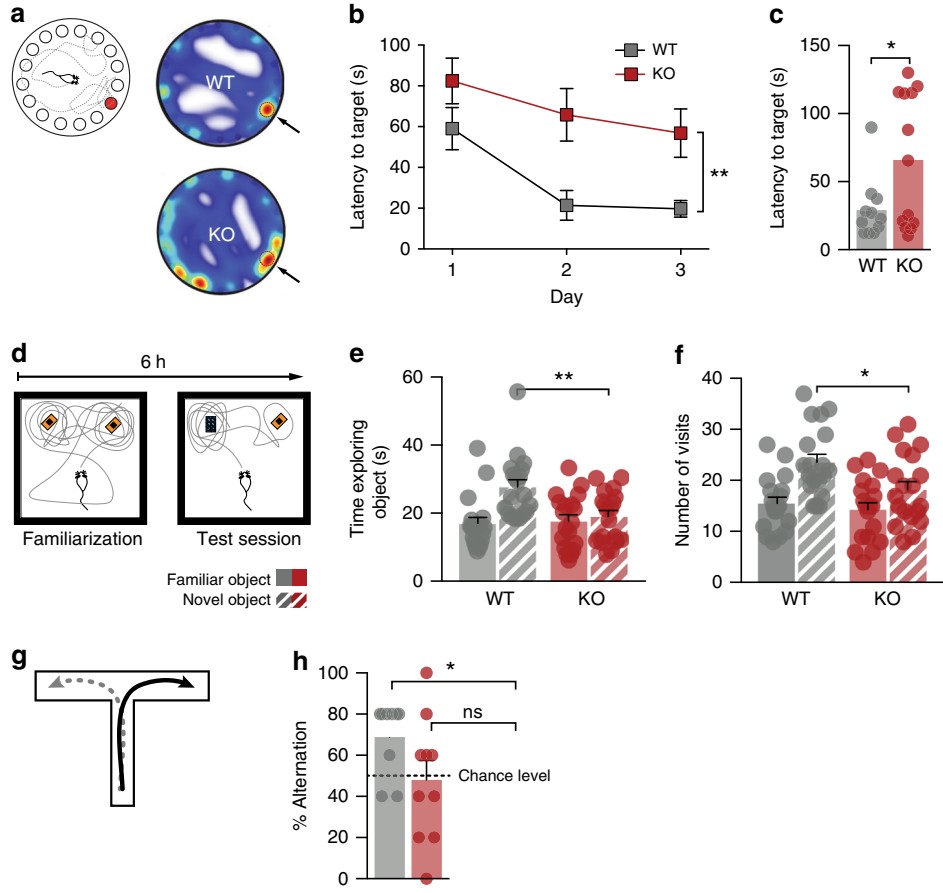

**Fig. 7** Memory and learning impairments in *Gprasp2*$^{-/y}$ mice. **a** Schematic diagram of a Barnes Maze test (left) and representative heat map analysis from probe trials of *Gprasp2*$^{-/y}$ mice and WT littermate controls (right). **b**, **c** *Gprasp2*$^{-/y}$ mice display increased latency to reach target hole during training sessions (**b**) and increased latency to reach target region on the probe day (**c**); WT $n = 11$, KO $n = 13$; repeated measures two-way ANOVA for (**b**) and two-tailed *t*-test for (**c**). **d** Schematic diagram of the novel object recognition task. **e**, **f** Reduced time (**e**) and number of exploratory (**f**) visits on a novel object test by *Gprasp2*$^{-/y}$ mice ; WT $n = 19$, KO $n = 19$; two-tailed *t*-test for (**e**) and two-tailed Mann–Whitney test for (**f**). **g** Schematic diagram of spontaneous alternation in a T-maze. **h** *Gprasp2*$^{-/y}$ mice display impaired alternation when compared with control mice; WT $n = 9$, KO $n = 10$; Wilcoxon signed-rank test against hypothetical value of chance alternation at 50%. All data are presented as means ± s.e.m. Statistical significance: *$p < 0.05$ and **$p < 0.01$

stimulation promoted a long-lasting and exacerbated induction of LTD in the CA3-CA1 circuit of *Gprasp2* mutant mice when compared with controls (WT: $87.47 ± 3.534\%$, $n = 11/7$ slice/mice; KO: $66.40 ± 6.338\%$, $n = 7/7$ slice/mice). This is consistent with perturbed activity of mGluR$_5$ receptors (Fig. 6g, h).

Taken together, our in vitro and in vivo results strongly suggest that GPRASP2 bidirectionally regulates mGluR$_5$ surface availability and that *Gprasp2* KO display abnormal mGluR signalling in hippocampal synapses.

**Gprasp2 mutant mice display ASD- and ID-like behaviours.** Considering the presence of *Gprasp2* mutations associated with ASD[38,39] and ID[36,37], we asked if *Gprasp2* mutant mice displayed behaviours reminiscent of neuropsychiatric disorders. We began by assessing motor function in the open field and in the rotarod test. Although no significant motor behaviour alterations were found between KO and WT littermates, there was a slight increase in the time spent in the centre of the open field (Supplementary Fig. 8a–d). While this could suggest altered basal anxiety levels, we found no significant changes in the light-dark box test (Supplementary Fig. 8e). To directly test for alterations in hippocampal function, we performed the Barnes maze test for learning and memory. In this test, *Gprasp2* mutant mice displayed an increased latency to find the target during training

trials, as well as an increased latency to reach the target region during the probe trial (Fig. 7a–c), when compared with littermate controls. In a novel object recognition test, *Gprasp2*$^{-/y}$ mutant mice showed a decrease in the time exploring and in the number of visits to the non-familiar object (Fig. 7d–f). Finally, we also found altered spontaneous alternation in the T-maze test for working memory (Fig. 7g, h). Together, our results suggest *Gprasp2*$^{-/y}$ mutant mice display learning and memory impairments that correlate with structural and functional hippocampal synaptic defects.

To study the impact of *Gprasp2*$^{-/y}$ in ASD-relevant behaviours we performed a modified version of a three-chamber social arena for voluntary initiation of social interaction and discrimination of social novelty[26]. In this test, *Gprasp2*$^{-/y}$ mice showed preference for a social partner 'S1' compared with an empty cage (Fig. 8a, b), however, their preference index for social interaction was significantly reduced (Fig. 8c). In a subsequent trial, a novel social partner ('S2') was introduced instead of an empty cage. In this case, *Gprasp2*$^{-/y}$ mice spent less time overall engaging in social interaction and also displayed a lower index of preference for the novel stimulus (Fig. 8d, e), however, the lack of preference for the 'S2' partner may result from both social recognition deficits compounded with the memory impairments displayed by *Gprasp2*$^{-/y}$ mice. During both trials, the total distance travelled

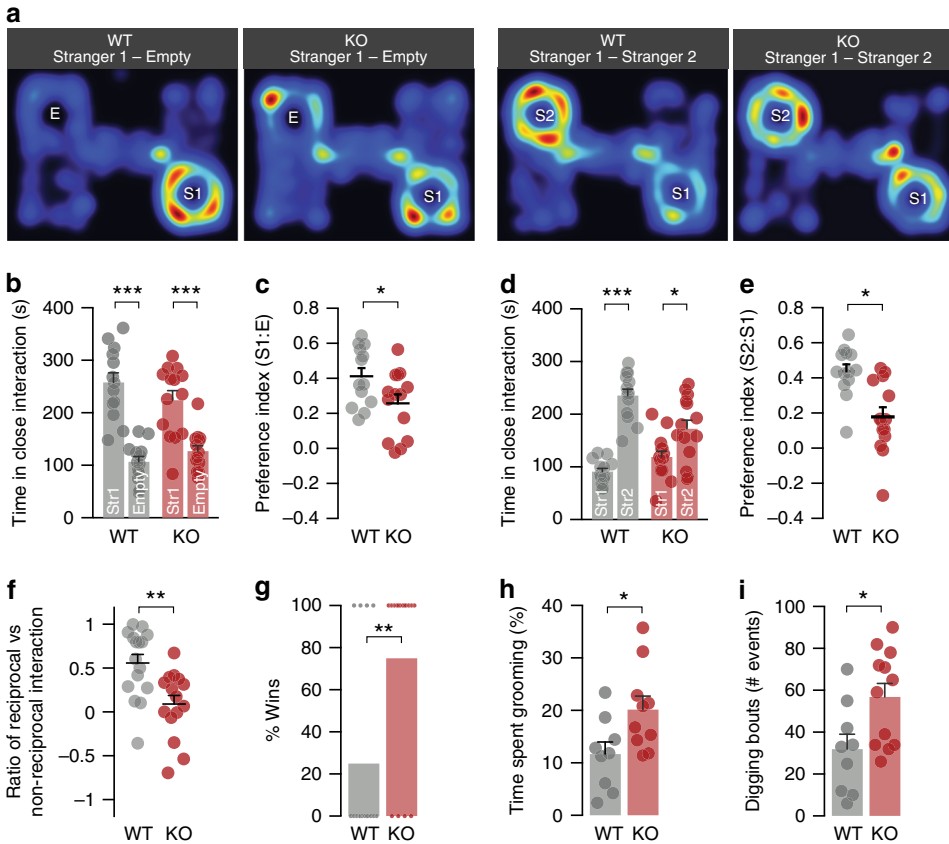

**Fig. 8** *Gprasp2−/y* mice display social and ASD-like behavioural alterations. **a** Representative heat map images from 'Stranger1–Empty' and 'Stranger1–Stranger2' from three-chamber social test trials for *Gprasp2−/y* mice and littermate controls. **b**, **c** *Gprasp2−/y* interact with 'Stranger' stimulus mice (**b**), but display reduced preference for social interaction when comparing with WT mice (**c**); WT n = 13; KO n = 14; two-tailed *t*-test. **d**, **e** *Gprasp2−/y* display reduced total time interacting with social partners (**c**) and reduced preference index for 'Stranger2' (**e**) when comparing to WT controls; WT n = 13; KO n = 14; two-tailed *t*-test. **f** In the social dyadic test, *Gprasp2−/y* show decreased ratio of reciprocated vs non-reciprocated social interactions against C57BL6 stimulus mice; WT *n* = 16, KO n = 15, two-tailed *t*-test. **g** In the tube test for social dominance, *Gprasp2−/y* mice displayed greater percentage of win trials against WT controls; WT *n* = 16, KO n = 16; chi-square test. **h**, **i** *Gprasp2−/y* mice display increased stereotypical behaviours measured as time spent grooming (**h**) and frequency of digging bouts (**i**) during a 30-min session; WT n = 9, KO n = 10; two-tailed *t*-test for (**h**) and two-tailed Mann–Whitney test for (**i**). All data are presented as means ± s.e.m. Statistical significance: *p < 0.05, **p < 0.01 and ***p < 0.001

by the mutant mice was significantly greater than controls (Supplementary Fig. 9a–b). In a social dyadic test, *Gprasp2* mutant mice engaged in significantly more events of non-reciprocal social interaction (i.e. the stimulus mouse ignores or retreats from the target animal) when compared with littermate controls (Fig. 8f). In the tube test for social dominance, *Gprasp2−/y* mice displayed a greater probability of winning by forcing a retreat of the WT mice out of the tube (Fig. 8g). We also found that innate social behaviour of the mutant mice was perturbed in the nest building test (Supplementary Fig. 9c).

Finally, we assessed stereotypical behaviours and found that mutant mice spent an increased amount of time grooming, digging in the homecage (Fig. 8h, i) and scrabbling in the open field test (Supplementary Fig. 9d). Together with the above, our data indicate the presence of ID- and ASD-like dysfunctions in *Gprasp2* mutant mice.

## Discussion

Abnormal synaptic development and plasticity may underlie a common pathophysiology in various human cognitive disorders[45]. Therefore, a better understanding of the cellular partners that regulate spine maturation, synaptic function, plasticity and mGluR physiology has the potential to provide valuable knowledge into disease mechanisms and contribute to

the development of innovative therapies. We created a novel mouse model to study the role of GPRASP2 and found that its deletion leads to enhanced DHPG-mediated plasticity, weakened synaptic transmission, reduced number of mature spines in hippocampal neurons and altered PSD ultrastructure. In addition, the behavioural phenotype of the knockout mice is reminiscent of both ASD and ID, which supports the genetic data implicating *GPRASP2* in human neurodevelopmental disorders. Although our work is centred on the interaction and regulation of mGluR5, *Gprasp2* could also have an impact on additional GPCRs. Indeed, not only have GPRASPs been proposed to potentially regulate more than one type of receptor, the promiscuous interaction between mGluRs and other GPCRs implies that GPRASP2 could modulate other receptors directly or indirectly[46]. Regardless, our experimental evidence clearly shows that the depletion of *Gprasp2* strongly impacts synaptic physiology and mGluR function.

Interestingly, *Gprasp1* knockout mice display behavioural dysfunctions linked to striatal and dopamine function[47,48], whereas the high expression levels of *Gprasp2* in the striatum is transient. Moreover, the physiological importance of *Gprasp1* and *Gprasp2* may be crucial in mammals, since a small genetic deletion containing both these genes produces neonatal lethality in homozygous animals and severe developmental complications in heterozygous progeny[49].

At their core, behavioural and cognitive dysfunctions will most likely arise from multi-mode alterations across a combination of circuits. Hence, it is tempting to speculate on the severity of the behavioural phenotypes displayed by *Gprasp2* mutant mice and the brain regions where this gene is most highly expressed. Particularly, juvenile animals strongly express *Gprasp2* in the hippocampal formation but also across various hypothalamic nuclei. Future studies should aim at dissecting circuit-specific dysfunction arising from *Gprasp2* deletion.

More broadly, our data adds to the evidence that mGluR activity is strongly implicated in neurodevelopmental disabilities and highlights the view that modulating neuronal endocytic partners may open new therapeutic avenues[31]. Our work also provides mechanistic insight into the endogenous regulation of mGluRs by GPRASP2 and supports the role of pathological mutations in *GPRASP2* leading to ASD- and ID-linked behaviours.

## Methods

**Generation of *Gprasp2*−/y mice**. *Gprasp2* mutant mice were generated by deleting exon 7 (the single protein-coding exon in the mouse *Gprasp2* gene) using cre/lox recombination. The targeting vector was introduced via homologous recombination in R1 ES cells using standard gene targeting methods and chimeric males were generated via blastocyst injection of positive ES cells[26,50]. Germline deletion was achieved via crossing *Gprasp2* cKO mice with β-actin Cre mouse line (Jackson Lab stock #019099) to generate F2 null offspring. *Gprasp2* mutant mice were viable and born at the expected Mendelian ratio. Genotypes were determined by PCR from mouse tail DNA: primer F1 (GAGCTCTTCCCCTCAGCATTAC) and primer R1 (GTGCCCAGTCATAGCCGAATAG) for the WT allele (643 base pairs) and primer F1 (GAGCTCTTCCCCTCAGCATTAC) and primer R2 (GCCCGAGAGGAAGATTTAGTTTC) for the mutant allele (730 base pairs).

**Mice**. Mouse cages were maintained at a constant temperature (22 °C) and humidity (60%), under a 12 h light/dark cycle (lights on from 7 am to 7 pm) in an individual cage ventilation system. Animals were allowed access to water and food ad libitum. Male animals, ages between 6 and 10 weeks, were used in the experiments performed in this study, unless otherwise noted. Tests were conducted from 9 am to 5 pm. Maintenance and handling of the animals was performed in compliance with all relevant ethical regulations for animal testing and research, including the guidelines of the Animals Use and Care Guidelines issued by FELASA and European Directives on Animal Welfare. All experiments with mice were carried out under animal testing research protocols approved by ORBEA (Institutional Animal Welfare Body of the University of Coimbra/CNC, reference number 127/2016) and DGAV (Portuguese Regulatory Agency, reference number 0421/2016). All behavioural tests and quantifications were performed by trained experimentalists blinded to animal genotype.

**In situ Hybridization**. mRNA in situ hybridization was performed as previously described[26]. Briefly, 15 μm cryosections from freshly frozen P5, P15 and 12-week-old mouse brain tissue were analysed using digoxigenin (DIG)-labelled probes against mouse GPRASP2 cDNA (GenBank Accession NM_001359371.1; region 676–1511 bp cloned into pBlueScript). The hybridization signal was detected using an alkaline phosphatase (AP)-conjugated anti-DIG antibody (Roche) and developed using 5-bromo-4-cloro-indolylphosphate/nitroblue tetrazolium (BCIP/NBT; Roche). Images were collected on a Zeiss Axioskop 2 Plus (Carl Zeiss, Thornwood, USA) with ×5 objective using a Zeiss Axiocam digital camera (Carl Zeiss).

**Open field**. Open field consisted of an opaque arena (40 × 40 × 30 cm) and mice were automatically video-tracked using Ethovision XT (Noldus, Netherlands). Mice were placed at the corner of the apparatus and locomotor behaviour was recorded for 1 h. Indirect and homogeneous illumination of the room was provided by white LED lamps at 100 lx. Time spent in the centre zone (15 cm × 15 cm) and distance travelled in the centre was evaluated.

**Rotarod performance**. Motor coordination was assessed in an accelerating rotarod test (4–40 r.p.m.). Animals were introduced in the apparatus (Med Associates) and the latency to fall was determined. Animals were given three successive trials in a single day for 3 days with an inter-trial interval of 10 min.

**Three-chamber social interaction test**. The three-chamber arena was from Stoelting (Stoelting, Ireland). *Gprasp2* KO and WT littermates were tested for voluntary social interaction as previously described[26]. The assay consisted of three sessions: the first session began with a 20-min habituation period during which the subject mouse freely explored all three chambers; next, the mouse was confined to

the centre chamber and an empty wire cage (Empty—'E') and a cage with an unfamiliar mouse (Stranger 1—'S1') were introduced to the side-chambers; in the second session, the subject mouse was then allowed to freely explore all three chambers for 10 min. Following the 10-min session, the animal remained in the chamber for an extra 10 min (post-test) to better acquire the identification cues from 'S1' animal. Before the third and last session, the subject mouse was gently guided to the centre chamber while the empty wire cage was replaced with a caged WT stimulus mouse (Stranger 2—'S2'). In the last session, the subject mouse was then left explored all three chambers for 10 min. Stimulus mice were males of the same age and previously habituated to the wire cages. The positions of the empty cage and 'S1' were alternated between tests. No position bias was observed. Time spent in close proximity, distance travelled, and heat maps were calculated using the automated software Ethovison XT (Noldus, Netherland). Preference index for each animal was calculated as $\frac{(S1-E)}{(S1+E)}$ or as $\frac{(S2-S1)}{(S1+S2)}$; where 'S1' is the time spent in close proximity with the stranger animal 1, 'S2' is the time spent in close proximity with the stranger animal 2 and 'E' is the time spent in close proximity with the empty cage.

**Dyadic social interaction test**. Mice were tested for reciprocal social interaction, as previously described[26]. Age-, sex- and weight-matched C57BL/6 males unfamiliar with the tested *Gprasp2*−/y mice and WT littermates were used as stimulus partners and their paws marked one week prior to the experiment. The test was performed in an open arena (40 × 40 × 30 cm) filled with fresh bedding. Illumination on the arena floor was kept at 100 lx during the test and the chamber was cleaned with 70% ethanol in between trials. The target mouse was removed from the homecage and placed on one side of the chamber and separated by a solid partition from the matched partner. After the 10-min acclimatization period, the barrier was removed and social interactions were recorded for 30 min. Social interactions were divided into two different categories, First, reciprocated or bidirectional social interaction is when the target mouse (WT/KO) engaged with the stimulus and the latter reciprocate by allogrooming or sniffing. On the other hand, non-reciprocated interaction is considered when the target mouse (WT/KO) initiated the social approach, but the stimulus did not reciprocate by ignoring or turning away from the test mouse. Trials involving animals engaged in fighting for more than 30 s without interruption were terminated. Quantification of these behaviours were scored manually by observers blinded to the genotype of the animals using the Observer XT 9 software (Noldus, Netherland).

**Nest building test**. Mice were individually housed in a new standard homecage (20 × 26 × 13 cm) with corn bedding and without environmental enrichment. In each cage, a single nestlet was added at 5 pm and recorded 16 h later. Nest quality produced by each mouse was assessed by at least three observers blinded to the experimental conditions following a 5-point rating scale[51].

**Novel object recognition**. The experimental arena was a white opaque arena (40 × 40 × 30 cm) and the test was performed with objects described before[52]. Habituation was done by exposing the animal to the arena for 10 min. The following day, in the familiarization session, mice were placed in the arena in the presence of two identical objects for 10 min. After a retention interval of 6 h, mice were again introduced in the arena with one familiar and one novel object. Mice were allowed to explore for 10 min. The objects chosen for this experiment were a 25-mL tissue culture flask filled with sand and a plastic Lego, both approximately the same height and weight. The duration of time mice spent exploring each object (familiar object vs novel object) was recorded by a trained observer, blind to the genotype using Observer XT 9 (Noldus, Netherlands).

**Dark-light emergence test**. This test consisted in a modified open field arena divided into two chambers with an entrance between the two parts. Mice were placed into the dark side of the two-chamber apparatus and were given 10 min to freely explore the arena while the illuminated side was kept under 400 lux. Mice were then filmed with a camera positioned overhead and the time spent on light side was quantified. The number of transitions and latency to first enter the light were analysed manually using the Observer XT 9 (Noldus, Netherlands).

**Tube test**. The tube test was performed in a transparent plexiglass tube, 33 cm long with an inner diameter of 3 cm. Acrylic ramps were placed to allow the animals access and retreat back from the tube. Testing started by introducing two different age-matched subjects to the edges of the tube. Testing ended as soon as one of the subjects had all paws outside of the tube for at least 4 s. All animals were weighed before each round and weight matched as closely as possible.

**T-maze test**. The apparatus consisted of a T-shaped maze (45.5 × 5 cm) elevated from the floor (60 cm). Fresh bedding was added to the maze before testing the animals. The mice were placed in the initial part of the stem and allowed to explore the maze. After the animal entered one of the arms, a sliding door was placed in the initial part of the arm chosen, allowing the mouse to explore the chosen arm. After a 30 s retention period, the animal was gently removed and returned to the homecage. Next, the animal was returned to the start arm and a second run was

initiated. Directions of choice was recorded for each mouse and the percentage of alternation obtained. Five trials (two runs each) were conducted in the space of two consecutive days (three on the first day and two on the second). The floor in the maze was illuminated at 15–20 lux.

**Barnes maze**. The Barnes maze test was performed to assess spatial memory and learning. The apparatus consisted in a white circular platform (122 cm diameter), elevated 92 cm from the floor, with 20 equally spaced holes (closed) with a diameter of 4.4 cm. An escape box was placed under one of the holes representing the target location. The spatial location of the target hole with respect to visual room cues was consistent between trials. Three training sessions of 2 min per day were performed. Twenty-four hours after the last training session, a probe trial was performed without the escape box to assess spatial memory. Trials were digitally recorded and analysed using the automated software Ethovision XT (Noldus).

**Repetitive behaviours quantification**. Animals were placed in a separate standard cage with bedding and recorded for 30 min. Quantified behaviours included self-grooming and digging and were manually scored by an observer blinded to the genotype of the mice. Self-grooming was defined as scratching of face, head or body with the two forelimbs, or licking body parts. Upright scrabbling/jumping was analysed during the open field test. Upright scrabbling was scored as fast, rhythmic movement of the forepaws, against the open field wall arena while the mouse is standing in an upright position. Quantification was performed off-line using the Observer software (Noldus Information Technologies).

**Electron microscopy**. Seven-week-old mice were deeply anesthetized with iso-flurane (IsoVet) and transcardially perfused with PBS (pH 7.4) followed by ice-cold 4% paraformaldehyde (PFA) in phosphate buffer (PBS; pH 7.4). Brains were removed, the hippocampus dissected and post-fixed overnight (o.n) in PFA 4%, then transferred into a 2.5% glutaraldehyde solution in 0.1 M sodium cacodylate buffer (pH 7.2) and kept at 4 °C o.n. The tissues were then rinsed in cacodylate buffer and post-fixed with 1% osmium tetroxide for 1 h. After rinsing in buffer and distilled water, 1% aqueous uranyl-acetate was added to the tissues, in the dark, during 1 h for contrast enhancement. Following rinsing in distilled water, samples were dehydrated in a graded acetone series (70–100%) and then impregnated and included in Epoxy resin (Fluka Analytical). Ultrathin sections (70 nm) were mounted on copper grids and observations were carried out on a FEI-Tecnai G2 Spirit Bio Twin at 100 kV. PSD measurements were performed using ImageJ (NIH, Bethesda, Maryland) by an observer blinded to the genotype of the samples.

**In vivo morphology of neurons and spines**. To achieve sparse, Golgi-like labelling of neurons in the CNS, we performed injections in the tail vein of 4-week-old animals with 5 μL of AAV9.Syn.eGFP.WPRE.bGH at a titre of $8.88 \times 10^{12}$ (Penn Vector Core, University of Pennsylvania, PA) diluted in sterile PBS to a final volume of 100 μL. Six weeks post-injection, animals were sacrificed, the brain collected and processed for neuronal imaging. Briefly, mice were anesthetized with isoflurane and perfused transcardially with PBS followed by 4% PFA in PBS, pH 7.4. Whole-brain was dissected and post-fixed in 4% PFA for 24 h, followed by transfer to a 30% sucrose solution in PBS. Serial coronal sections of 100 μm were collected using a vibratome (Leica VT1200s, Leica Microsystems, USA) and mounted in gelatinized slides using Vectashield with DAPI (Vector Laboratories) as mounting medium. Slides were stored at 4 °C protected from light until further analysis. Images of pyramidal neurons from the CA1 region of the hippocampus were acquired in a LSM 710 Confocal microscope (Zeiss, Germany) with a Plan Apochromat 20x/0.8 DICII lens. Each image consisted of a stack of images taken through the z-plane of the section. Confocal microscope settings were kept the same for all scans in each experiment. Neurons expressing GFP were chosen randomly for quantification from at least four different sections containing the region of interest and at least six neurons were acquired per animal. Neuronal tracing reconstruction was performed using Neurolucida (MBF Bioscience). Spines on secondary dendrites of hippocampal neurons were acquired using a LSM 710 Confocal microscope (Zeiss, Germany) with a Plan Apochromat 63x/1.4 NA oil objective. Spine density and size analysis was performed using Neurolucida (MBF Bioscience). Each spine was included in one of two categories: immature spines which included filopodia (spines without a defined head) and mature spines which included stubby (spines without a defined neck); thin spines (with a neck and head diameter smaller than double the width of the neck) and mushroom spines (with a neck and head diameter larger than double the width of the neck). Sholl analysis was performed using the Neuroexplorer software (MBF Bioscience). Experiments were performed blind to animal genotype during both image acquisition and image analysis.

**Preparation of brain slices for electrophysiology**. Acute hippocampal slices were prepared from P15 to P20 *Gprasp2* KO and WT littermates by an experimentalist blinded to the genotype of the animals. The osmolarity of all solutions was adjusted to 300–310 mOsm except for the sucrose-enriched buffer for whole-cell patch clamp (SB-PC; osmolarity 330–340 mOsm) and the cesium-based internal solution (Cs-Int; osmolarity was adjusted to 295–298 mOsm with CsMeSO₃). The pH of all

solutions was adjusted to 7.36 with HCl, except for the Cs-Int which was adjusted with CsOH.

**Field recordings and DHPG-mediated long-term depression**. Before brain dissection, mice were deeply anesthetized with isoflurane and perfused with an oxygenated (95%:5% $O_2$:$CO_2$ mix) sucrose-enriched buffer (SB) containing (in mM): sucrose 198.86, KCl 2.55, NaHCO₃ 25, NaH₂PO₄.2H₂O 1.09, glucose 25.03, MgSO₄ 2.5 and CaCl₂ 0.5. The brain was quickly removed and glued to a vibratome support filled with ice-cold, oxygenated SB. Sagittal hippocampal slices of 300 μm were obtained using a vibratome (Leica VT1200s, Leica Microsystems, USA) and immediately recovered at 32 °C for 30 min in oxygenated artificial cerebrospinal fluid (aCSF) containing (in mM): NaCl 130.9, KCl 2.55, NaHCO₃ 24.05, NaH₂-PO₄.2H₂O 1.09, glucose 12.49, MgSO₄ 0.5 and CaCl₂ 2.

Before recording, hippocampal slices were placed for at least 1 h at RT in oxygenated aCSF. Slices where then moved to the recording chamber and perfused with oxygenated aCSF (2–3 mL/min) at 25 °C. The hippocampus was visualized with an Axio Examiner.D1 microscope (Zeiss, Germany) equipped with a Q-capture Pro7 camera (QImaging, Canada) and fEPSPs were recorded in CA1 stratum radiatum using a borosilicate glass (Science Products) recording electrode filled with aCSF (2–4 MΩ) and placed at the depth in the slice that gave the largest signal amplitude followed by a stable signal response for 10 min. Evoked responses were obtained by stimulating the Schaffer collaterals at 0.05 Hz with a concentric bipolar stimulating electrode (0.2 ms stimulus; Bowdoin, ME, USA) connected to a stimulator Digitimer model DS3 (Digitimer, UK). The current applied was calculated as 50% of the maximal response in an input–output curve starting at 20 μA with 10 μA increments. The LTD protocol consisted in attaining a stable baseline for 20 min followed by application of 50 μM (S)-3,5-DHPG (Tocris, Bristol, UK) for 5 min and performing continuous recording for an additional 55 min. Field potentials were filtered at 0.1 Hz–1 kHz, and digitized at 10 kHz. Basal synaptic transmission was assessed with the input–output curve and via paired-pulse facilitation (PPF) according to protocols described in the literature[14]. Briefly, facilitation was assessed by applying two consecutive pulses separated by 20, 30, 50, 100, 200, 300 and 500 ms inter-stimulus intervals and plotting the ratio of the fEPSP slope of stimulus 2 to stimulus 1.

**Whole-cell patch clamp**. For CA1 pyramidal neuron whole-cell recordings, the brains and slices were handled as described above, except for the cutting and recording solutions. In this case, the SB-PC used for both perfusion and slicing contained (in mM): sucrose 75, NaCl 86.93, KCl 2.55, NaHCO₃ 25, NaH₂-PO₄.2H₂O 1.09, glucose 25.03, MgCl₂ 1.75 and CaCl₂ 0.5. Simultaneously, the aCSF-PC used for slice recovery and recording contained (in mM): NaCl 125.09, KCl 2.55, NaHCO₃ 25, NaH₂PO₄.2H₂O 1.09, glucose 25.03, MgSO₄ 0.5 and CaCl₂ 2. Following the previously described recovery periods, CA1 pyramidal neurons were identified under infrared-differential interference contrast (IR-DIC) visualization. Cells were patched with borosilicate glass recording electrodes (3–5 MΩ; Science Products) filed with a Cs-Int solution containing (in mM): CsMeSO₃ 115, CsCl 20, MgCl₂.6H₂O 2.5, HEPES 10, EGTA 0.6, Na-phosphocreatine 10, ATP sodium salt 4 and GTP sodium salt 0.4. Three minute recordings were performed at 30 °C in oxygenated aCSF-PC in the presence of TTX (10 μM), bicuculline (40 μM) and D-APV (20 μM) to isolate AMPAR-mediated mEPSC. CA1 neurons were voltage-clamped at −80 mV to amplify the smallest spontaneous miniature synaptic events that might otherwise escape detection, as previously described[26]. Criteria for acceptance of cells was determined as a stable Ra under 25 MΩ. Recordings were filtered at 2 kH and digitized at 20 kHz.

**Data acquisition and analysis**. Data were acquired with a Multiclamp 700B amplifier and Digidata 1550A (Molecular Devices Corporation) and analysed using Clampfit 10.7 software (Axon Instruments). For the LTD protocol, each fEPSP datapoint corresponds to the average of three slopes (one slope every 20 s) and was normalized to the mean of the 20-min baseline. Rise and decay time of mEPSC were determined as the time interval to get from 20 to 80% of the maximal amplitude and from 90% to 10%, respectively, as previously described[53].

**Tissue collection and biochemical analysis**. Animals (P15–P20) were anesthetised with isoflurane (IsoVet) and euthanised by decapitation. The hippocampus and other brain regions were dissected on ice in a dissection microscope. Until processing tissue was stored at −80 °C. Synaptosomal plasma membrane was purified as described previously[26]. Each sample contained material from a pool of 4–5 animals (WT or KO). Protein quantification was performed using the BCA protein assay kit (Pierce BCA protein assay Kit, Thermo Scientific), following the manufacturer's instructions. Protein sample (10 μg) was denatured with 4x Laemmli sample buffer (Bio-Rad) and 10% β-mercaptoethanol. Samples were incubated at 95 °C for 5 min before western bloting.

**qRT-PCR of GPRASP family members and primer list**. Animals (P20) were anesthetised with isoflurane (IsoVet) and euthanised by decapitation and brain tissue was flash frozen until processed. Briefly, total RNA was extracted using the NucleoSpin RNA kit (Macherey Nagel) according to the instructions of the supplier. Complementary DNA (cDNA) was synthesized from 10 ng of total extracted

RNA using the Fluidigm Reverse Transcription MasterMix kit (Fluidigm), following the instructions from the supplier. The resulting cDNA was subjected to quantitative PCR analysis using a 96.96 Dynamic Array Integrated Fluidic Circuit and the Fast Gene Expression Analysis using EvaGreen on a Biomark HD System. Beta-2-Microgobulin (B2M) was used as an internal control for all samples. PCR primer sequences used were as follows (gene name: 5′-3′ forward primer/reverse primer):

*Gprasp1*: CCAGGCAAAGCGCTGAAAATA/
GATTTGTGTCCTAACCTTGGGTC;

*Gprasp2*: TGTGAAGGTCGCCTGCCG/TCCAGTACCAATGAGACTCCTA;

*Gprasp3*: AGGGTCTAAGGGAAAGGTAGTTG/
CGTGTGGATCTAGCAAACTTGT;

*Gprasp4*: ACTGGAGTGGACACGAAGTC/
AGCACCAGCCATATCATCATTTT;

*Gprasp6*: AAGGGCTTCTCCTAATTCAGACG/
GCAGCATTATTACCCAGAGCAA;

*Gprasp7*: CTGGTGCCTGCTACTGTGTAT/
CCCCTACCCCAACATTAGTCT;

*Gprasp8*: TGCTACTGTATCTACCGGCTG/
GGTTAGGTCTTCTGCGGATCG;

*Gprasp9*: CTGGAATCAGTAGTCATGCCTTC/
AGTCTGGGCTATCATTGGAGAT;

*Gprasp10*: TGGGAAGAAGTGAGGGGAAC/
GTCGAGCCATTGCTGTGAAAT

**Western blotting**. Equal amounts of proteins were resolved by SDS-PAGE in 8–10% polyacrylamide gels. Proteins were transferred to PVDF membranes Amersham Hybond (GE Healthcare Life Sciences) blocked at room temperature (RT) for 1 h in ×TBS with 5% milk + 0.1% Tween-20 (TBS-T). Membranes were incubated with primary antibodies at 4 °C overnight before washing in TBS-T and incubated with secondary antibodies (1:10,000) of horseradish peroxidase (HRP)-conjugated donkey (anti-mouse, anti-rabbit, Jackson ImmunoResearch) for 120 min at room temperature. Following three time washes in TBS-T, the membranes were incubated with ECL western blot substrate (GE Healthcare Life Sciences). Signal was visualized on a Storm 860 Gel and Blot Imaging System (GE Healthcare Life Sciences). Antibodies used in western blotting experiments were the following: anti-PSD-95 (6G6-1C9, 1:1000; Cell Signaling Technology), anti-PSD-93 (N18/ 275-284, 1:1000 NeuroMab), anti-CAMK2a (6G9, 1:1000, Sigma-Aldrich), GluA1 (AB1504, 1:1000, Millipore), GluA2 (MAB397, 1:1000, Millipore) and anti-GPRASP2 (ab129417, 1:100; Abcam or 12159-1-AP, 1:1000; Proteintech). When indicated, anti-β-tubulin (T7816, 1:20,000; Sigma-Aldrich) or anti-β-actin (A5441, 1:5000; Sigma-Aldrich) antibodies were used as loading controls. Uncropped western blots are shown in Supplementary Fig. 10.

**HT-22 cell culture co-immunoprecipitation**. For co-immunoprecipitation experiments, HT-22 cells were co-transfected with 1 µg of pmGluR₅ₑ and pGPRASP2-GFP using polyethyleneimine and allowed to express for 48 h. Cells were harvested in control conditions or after stimulation for 5- or 30-min with 100 µM DHPG. Harvesting was performed in TEEN Buffer (25 mM Tris pH 7.4, 1 mM EGTA, 1 mM EDTA, 150 mM NaCl, 1% Triton), supplemented with 1 mM DTT, 1 µg/ml CLAP and 0.2 mM PMSF. Samples were sonicated for 45 s and then centrifuged at 15,000 × *g* for 10 min. Next, 800 µg of protein extract were incubated (2 µg/µL) with either anti-GPRASP2 antibody (3 µg of 12159-1-AP, Proteintech) or IgG (3 µg of control IgG, Millipore), overnight at 4 °C. Sepharose A beads were added (80 µl) and the mixture incubated for 2 h at 4 °C. Beads were washed twice with TEEN Buffer + 1% DDM and then twice with TEEN Buffer. Sample buffer was used to eluted protein from the beads and samples were boiled prior to western blotting.

**GPRASP2 co-localization with endocytic pathway markers**. Cell culture of HT-22 were used and transfected with 1.5 µg of pGPRASP2-GFP and 1.5 µg of either Lamp1-RFP, mRFP-Rab5, mRFP-Rab7 or mRFP-CLC. Lamp1-RFP was a gift from Walther Mothes, Addgene #1817[54], mRFP-Rab5, mRFP-Rab7 and mCLC were a gift from Ari Helenius, Addgene #14436, #14437 and #14435[55,56]. Culture imaging was performed in an LSM 710 Confocal microscope (Zeiss, Germany) with a Plan Apochromat 63x/1.4 NA oil objective. Data analysis and co-localization was performed using Fiji[57].

**Hippocampal neuron primary cultures**. Low-density primary cultures of rat hippocampal neurons were prepared from the hippocampi of E18 Wistar rat embryos, as previously described[53]. Briefly, after dissection the tissue was treated for 15 min at 37 °C with trypsin (0.06%, Gibco Invitrogen) in Ca²⁺- and Mg²⁺-free Hank's balanced salt solution [HBSS (in mM): KCl 5.36, KH₂PO₄ 0.44, NaCl 137, NaHCO₃ 4.16, Na₂HPO₄.2H₂O 0.34, glucose 5, sodium pyruvate 1, HEPES 10 and 0.001% phenol red]. Cells were then washed six times in HBSS and mechanically dissociated. Cells were plated in neuronal plating medium (MEM supplemented with 10% horse serum, 0.6% glucose and 1 mM pyruvic acid) onto poly-D-lysine-coated coverslips in 60 mm culture dishes, at a final density of 3 × 10⁵ cells/dish. After 2–4 h, coverslips were flipped over an astroglial feeder layer in Neurobasal

medium [supplemented with SM1 neuronal supplement (StemCell Technologies, Grenoble, France), 25 µM glutamate, 0.5 mM glutamine and 0.12 mg/ml gentamycin]. Wax dots on the neuronal side of the coverslips allowed the physical separation of neurons from the glia, despite neurons growing face down over the feeder layer. To further prevent glia overgrowth, neuron cultures were treated with 5 µM cytosine arabinoside after 3 DIV. All cultures were maintained at 37 °C in a humidified incubator of 5% CO₂/95% air. Neurons were transfected at (1) DIV7 and imaged at DIV15 for GPRASP2 overexpression experiments labelling PSD-95 and VGLUT1; (2) DIV11 and imaged at DIV17 (to attain more mature neurons) in spine and Sholl analysis and (3) DIV10 and imaged at DIV15 for GPRASP2 knockdown experiments, due to higher toxicity of the shRNA construct. The rescue construct used to express GPRASP2 coded the fusion protein GPRASP2-GFP and was cloned in-house (see below). The shRNA constructs targeting Rat GPRASP2 mRNA are described below. The experiments using MPEP incubation proceeded as described above, but MPEP at a concentration of 20 µM was added each day from DIV10 to DIV14.

**Plasmid constructions**. The full-length mouse coding sequence for GPRASP2 (aa 584-919) was amplified by PCR from BAC DNA clones (RP23-160E, RP23-250G; Children's Hospital Oakland Research Institute, USA) and cloned into pEGFP-C1 (Clontech, USA) using restriction enzymes (XhoI, KpnI). The cloning was confirmed by restriction digestion analysis and sequencing. In order to knockdown GPRASP2, four unique short hairpin RNA (shRNA and shRNA II) sequences that target rat GPRASP2 and one non-targeting sequence (negative control) were tested (Catalogue# 336311 KR54520G; SureSilencing, Qiagen). Each plasmid vector expressed a shRNA under the control of a U1 promoter and contained a GFP reporter gene. The shRNA nucleotide sequences tested were (1) shRNA: 5′-AAGCCCAGGTCCAAACAAGAT-3′; shRNA II: 5′-ATTCGTGGGTCTCTT-TAATAT-3′; Scramble sequence: 5′-GGAATCTCATTCGATGCATAC-3′. The full-length mGluR₅ₑ was amplified by PCR from a mouse brain cDNA library and cloned into pCerulean C1 with EcoRI/SalI restriction sites; in a next cloning step Cerulean was removed using PCR-mediated plasmid deletion to create Cerulean and Cerulean-free pmGluR₅ₑ. Cloning was confirmed by restriction digestion analysis and sequencing.

**Immunocytochemistry**. Neurons were fixed for 15 min in 4% sucrose and 4% PFA in PBS (in mM: NaCl 137, KCl 2.7, KH₂PO₄ 1.8 and Na₂HPO₄.2H₂O 10, pH 7.4) at RT, and permeabilized with PBS 0.25% (v/v) Triton X-100 for 5 min, at 4 °C. Neurons were then incubated in 10% (w/v) BSA in PBS for 30 min at 37 °C to block non-specific staining and incubated with the appropriate primary antibodies diluted in 3% (w/v) BSA in PBS (overnight at 4 °C). The following primary antibodies and dilutions were used: anti-MAP2 (ab5392, 1:5000, Abcam), anti-PSD-95 (MA1-045, 1:750, Thermo Fischer), anti-mGluR₁/₅ (75-116, 1:200, NeuroMab), anti-VGLUT1 (AB5905, 1:1000, Millipore). After washing six times in PBS, cells were incubated with the respective secondary antibodies diluted in 3% (w/v) BSA in PBS (45 min at 37 °C): Alexa 568-conjugated anti-mouse (A-11004, 1:500, Molecular Probes), Alexa 488-conjugated anti-rabbit (A11008, 1:500, Molecular Probes), AMCA-conjugated anti-chicken (103-155-155, 1:200, Jackson ImmunoResearch) and Alexa 647-conjugated anti-guinea pig (A-21450, 1:500, Molecular Probes). The coverslips were mounted using fluorescent mounting medium (Dako, no. S3023). For quantification of surface mGluR₅, neurons were incubated with 100 µM (S)-3,5-dihydroxyphenylglycine (DHPG; Tocris) or vehicle (dH₂O) for 30 min at 37 °C. Following washing with PBS, cells were re-incubated for 1 h at 37 °C to allow receptor internalization. Neurons were fixed and incubated o.n. with the mGluR₅ intracellular N-terminus antibody (AGC-007, 1:100, Alomone Labs). Next, as indicated above and for further staining of neuronal proteins, each coverslip was washed, permeabilized and incubated in 10% (w/v) BSA in PBS for 30 min at 37 °C to block non-specific staining. The remaining labelling procedure was performed as described above.

**Imaging and quantitative analysis of hippocampal neuron cultures**. Imaging was performed on an Axio-observer Z1 microscope using a Plan Apochromat 63x/ 1.4 NA oil objective. For each experiment, images in each channel were captured using the same exposure time across all fixed cells. Images were quantified using image analysis software FIJI (FIJI Is Just ImageJ) with custom made macros. For each neuron, two to three dendrites were chosen for analysis from the dendritic marker image and their length was measured using MAP2 staining. The staining signals were analysed after thresholding and recognizable clusters under those conditions were included in the analysis, and measured for cluster signal intensity, number of clusters, and average area of clusters for the selected region. Measurements were performed in two to five independent preparations with at least five cells per condition analysed for each preparation.

Neuron culture spine imaging was performed on an Axio-observer Z1 microscope using a Plan Apochromat 63x/1.4 NA oil objective. For analysis, spines were categorized into five different groups (stubby, mushroom, short, long and filopodia) based on the following cut-off values: stubby, no neck; mushroom, neck ≤ 0.5 µm and head > 0.5 µm; short, neck < 2 µm in length; long neck ≥ 2 µm in length and filopodia headless protrusion. Automated spine quantification was performed using NeuronStudio software (Mount Sinai School of Medicine). For

Sholl analysis, transfected neurons were chosen randomly for quantification and at least six neurons were acquired per condition. Images were acquired as described before using a Plan Apochromat 20x/0.8 DICII lens and traced in Neurolucida (MBF Biosciences) and quantification was performed using Neuroexplorer (MBF Bioscience).

**Statistical analysis**. Data is represented as mean values ± s.e.m. or as frequency distribution plots (as indicated in figure legend). Statistical analysis was performed using unpaired two-tailed Student $t$-test, two-tailed Mann–Whitney test, one-way or two-way ANOVA analysis followed by post hoc test indicated in figure legends or one-sample Chi-square test. Sample normality was tested using D'Agostino-Pearson normality test. Analysis were performed using Graphpad (Prism) or Matlab (Mathworks). Statistical significance was defined as $***p < 0.001$, $**p < 0.01$, $*p < 0.05$.

**Reporting summary**. Further information on experimental design is available in the Nature Research Reporting Summary linked to this article.

## Data and code availability

The data, computer scripts, protocols and biological materials included in this study are available from the corresponding author upon reasonable request. A reporting summary for this article is available as a Supplementary Information file.

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

## Acknowledgements
This research was supported by the Portuguese Foundation for Science and Technology (FCT) Investigator Programme IF/00812/2012, FCT grant POCI-01-0145-FEDER-016682, Marie Curie Career Integration Grant (618525), NARSAD Young Investigator Grant from the Brain & Behaviour Research Foundation (#20733) and Bial Foundation Grant (266/2016) to J.P. We thank the support from FEDER/COMPETE institutional funds POCI-01-0145-FEDER-007440, BrainHealth 2020 CENTRO-01-0145-FEDER-000008; fellowships SFRH/BD/51958/2012 (FCT PDBEB to M.E), SFRH/BPD/120611/2016 (FCT to J.R.G.), SFRH/BD/105878/2014 (FCT MIT-Portugal to M.J.C.) and NIMH grant R01MH097104 (to G.F.). We thank C. Semião and S. Freire for assistance with colony management and husbandry; M. Zuzarte and LABCAR (Faculty of Medicine, University of Coimbra) with electron microscopy sample preparation and imaging; L. Cortes and MICC for assistance with microscopy imaging; J. Valero with help designing macros in Fiji. The R1 ES cells used to generate *Gprasp2* mutant mice were a kind gift to J.P. from Dr. Andras Nagy (Mount Sinai Hospital). We thank S. Santos and T. Catarino for assistance with neuron cultures and all members of the J.P. lab for helpful discussion and comments.

## Author contributions
M.E., J.R.G., M.I.P., M.L., M.J.C., G.C., X.G., L.O.F., P.A.F. and D.W. participated in the execution and analysis of experiments. M.E., J.R.G., M.I.P., A.L. Cardoso, A.L. Carvalho, G.F. and J.P. interpreted the results. A.L. Carvalho and G.F. contributed with critical resources and reagents. M.E., J.R.G. and J.P. designed the experiments and wrote the paper. All authors edited and approved the manuscript.

## Additional information

**Competing interests:** The authors declare no competing interests.

