## [Peer Review File · Nature Communications]

Reviewers' comments:

Reviewer #1 (Remarks to the Author):

By using transgenic mice and primary neuronal cultures, this study shows the role of GPRASP2 in neuronal structural growth and brain function including dendritic arborization, synaptogenesis, synaptic plasticity, and ASD-related cognitive and social behavior. GPRASP2 is implicated in ASD and schizophrenia, but its neural function remains largely unknown. Therefore, this study is novel and of value to the field. However, the paper suffers from a major weakness due to a lack of detailed mechanistic investigation. It is unclear whether and how the multiple aspects of the observation are connected and how they contribute to ASD.

Major points

1. GPRASP2 is one member of the GPRASP gene family. How is it different from other members (in function, cellular expression, regulation etc)?
2. It seems arbitrary to hypothesize that GPRASP2 regulates mGluR5. How does GPRASP2 regulate mGluR5? Maybe by direct interaction? If so, Co-IP should be performed. Is the association specific to mGluR5 compared to other mGluRs and other GPCR?
3. Fig 1D, which brain region was used? Multiple brain regions including hippocampal and cortical tissue should be used to compare the expression pattern.
4. Fig 1I, changes at the distal region suggests that the alteration is limited to apical dendrite; so that a comparison of the basal vs. apical dendrites will be helpful. Is the change specific for hippocampal CA1 neurons? How about cortical neurons?
5. Fig 2, the mechanism underlying the reduction in mEPSC amplitude was not studied. At least the expression of AMPAR should be explored. Also, the study showed that the dendritic arborization was reduced, and therefore the total synapse number in a neuron is expected to be lower. However, mEPSC frequency was not changed. Why?
6. Fig 3, the density of mature synapse was reduced while that of the immature synapse showed no change. However, the overall synapse density showed no change. What causes the inconsistency?
7. Fig 4, there is no confirmation on the shRNA knockdown effect on GPRASP2 in neurons. Similarly, it is important to confirm that the GPRASP expression is recovered to the basal level (i.e., rescued). Because the rescue expression might actually lead to GPRASP2 overexpression.
8. What is the possible mechanism for the alterations in dendrite growth and spine formation? Via mGluR5 signaling?
9. Fig 6 AB, the effect on mGluR1 was not specifically studied (only studied in combination with mGluR5). What is the underlying mechanisms? Does the change in mGluR cluster result from alterations in total mGluR protein amount, trafficking or synapse maturation? In addition, a non-Group I mGluR member should be used as a negative control.
- 10 Fig 6G, there is basically no difference except for the small portion after 60min? Is that due to worsened cell conditions? How about LTP and LFS-induced LTD given the changes in hippocampal synaptic change and impaired memory?
11. Fig 8, the KO mice showed minor abnormalities in social interaction (Fig 8B). More significant change was detected in social novelty (with stranger 2), but impairments in memory may, at least partially, contribute to the change.

Reviewer #2 (Remarks to the Author):

In this manuscript, Peca et al. explore the function of Gprasp2, a little-studied GPCR sorting protein linked to ASD and ID. Overall, this study represents a broad and thorough characterization of a constitutive knock-out allele. Given the association of this molecule with human disease, as well as its potential ability to bind a wide range of GPCRs throughout the body and brain, I think this work would be of broad interest to the neuroscience field. There is a small literature regarding the function of this molecule and this work would make a significant contribution to that knowledge. The experiments represent standard approaches in the field and appear well executed, soundly interpreted and nicely illustrated. Overall, the statistical tests are typical for these assays.

Currently, there are a few issues that should be addressed to strengthen and round out this story:

Major Points:

1. the work in dissociated culture does not dramatically enhance the previous findings and raises far more issues than it solves. First, if the authors were interested in exploring the cell autonomous nature of the phenotype, why didn't they just use their conditional allele and the elegant AAV9 mosaic infection approach? This would allow us to continue to compare apples and apples, as opposed to now having to compare in vivo manipulations in an intact mouse brain with dissociated cultures (from another species) that are undergoing a new wave of synaptogenesis within the dish. Using this AAV9 approach, they could have also injected at different times to get at questions of compensation as well as distinct temporal roles for this protein (development vs mature function). As it stands, it creates the need to reconcile the specific effects on mature spines of the KO from the broad spine changing effects of the shRNA - what do these differences mean?, do they reflect developmental vs. maintenance phenotypes? Furthermore, this dendrite and spine density phenotype is always a potential problem when using shRNAs in hippocampal cultures (Alvarez et al. JNeuro. 2006), requiring very strong controls. Along these lines, the authors do use a scrambled shRNA as well as a WT replacement. However, the replacement results are difficult to interpret in light of the spine density GOF phenotype (Fig5) because you may just additively be getting rid of off-target effects. In fact, it does seem like the KD+rescue has lower levels of spines (Fig4E) than the O/E alone (Fig5E). alternatives to prove lack of off-target effects would be another KD showing the same phenotype or using the KD construct employed in Fig4 on the KO mouse background (and seeing no dendrite length/spine density phenotype). Overall, the strongest point to be taken from the in vitro work is that Gprasp2 can modulate surface levels of mGluR5. Even here though, it does seem that some of this is hard to interpret given that some of the manipulations also cause global changes in mGluR5 protein levels (Fig 6E,F). These concerns about cell autonomy and compensation are valid but could be addressed more strongly.

2. while mentioned in the text, one major issue here is the implication of mGluR specificity for a protein whose specificity towards its GPCR binding partners is not clear. While I believe the authors that this mutation can alter mGluR surface levels, the paper is written in a way to suggest that this is a likely explanation for much of what follows, both at the morphological and behavioral level. Because this link with mGluR signaling enhances the profile and translation relevance of the work, I would suggest the authors try to draw a more concrete link. To do this, what about taking the mutant and using either systemic or targeted negative modulators of mGluR5 signaling to attempt to ameliorate either a morphological (probably easier) or behavioral phenotype. If you have a strong hypothesis that the deficit is developmental in origin, perhaps just using this modulation during neuronal culturing and looking at morphology?

Minor Points:

1. Fig 1: I don't see much evidence for specificity of expression, either at P15 or at P5 in the striatum. This molecule looks widely expressed. Regional qPCR would be more convincing if one really wanted to show this.
2. Fig2E: bad trace example, shows the reduced amplitude but also has slower kinetics.
3. Fig3A: missing 'A' label.

4. There are numerous grammatical errors throughout the text that should be fixed.

Reviewer #3 (Remarks to the Author):

This is a timely and important study examining a mutant mouse model with a deletion of the GPCR interacting protein Gprasp2. Human mutations in this gene have been linked to autism and intellectual disability. In this study, the authors examine the neurological phenotypes associated with loss of Gprasp2 especially with respect to modulation of group 1 mGluRs, which are implicated in autism and ID.

This study is impressive, using multiple methodologies to examine biochemical, structural, electrophysiological and behavioural phenotypes in the Gprasp2 mutant mouse. The overall conclusions of the authors are that reduction of Gprasp2 leads to significant disruptions in synaptic structure and function in hippocampal neurons, and an exaggeration of mGluR-mediated plasticity that may be due to a change in the trafficking of mGlu5 receptors. This in turn leads to deficits in learning and social behaviour reminiscent of other autism/ID mouse models.

It is clear that the authors were rigorous in the experimental design and analysis for each part of this study, and this reviewer has no reservations for recommending it for publication. However, a few additions might strengthen the manuscript:

- 1.) The overexpression of mGlu5 at the surface suggests that the intracellular signalling downstream of this receptor should be enhanced. Have the authors tested whether this is the case? Do they believe that other functions of group 1 mGluRs might be exaggerated?
- 2.) Have the authors thought of testing whether negative allosteric modulators such as MTEP or CTEP could reverse phenotypes in the Gprasp2 mutant?
- 3.) Other GPCR interacting proteins have been implicated in autism and ID (ie, Homer1, arrestins). Is there a phenotypic similarity between these other models and the Gprasp2 mutant?
- 4.) Given the role of mGlu5 in anxiety, it is interesting that there appears to be no anxiety phenotype in the Gprasp2 mutant. Do the authors have any thoughts on why this might be the case?

Abnormal mGluR-mediated synaptic plasticity and autism-like behaviours in *Gprasp2* mutant mice

Mohamed Edfawy; Joana R. Guedes; Marta I. Pereira; Mariana Laranjo, Mário J. Carvalho; Xian Gao; Pedro A. Ferreira; Gladys Caldeira; Lara O. Franco; Dongqing Wang; Ana L. Cardoso; Guoping Feng; Ana L. Carvalho; João Peça

Reviewer #1

We are grateful to the reviewer for his/her careful reading of our manuscript, encouraging comments regarding novelty and value, and for suggesting critical experiments that have improved the manuscript. We have included new data that support the conclusions of the study and rephrased parts of the manuscript in order to clarify our arguments.

By using transgenic mice and primary neuronal cultures, this study shows the role of GPRASP2 in neuronal structural growth and brain function including dendritic arborization, synaptogenesis, synaptic plasticity, and ASD-related cognitive and social behavior. GPRASP2 is implicated in ASD and schizophrenia, but its neural function remains largely unknown. Therefore, this study is novel and of value to the field. However, the paper suffers from a major weakness due to a lack of detailed mechanistic investigation. It is unclear whether and how the multiple aspects of the observation are connected and how they contribute to ASD.

Major points

1. GPRASP2 is one member of the GPRASP gene family. How is it different from other members (in function, cellular expression, regulation etc)?

- i) The GPRASP family is composed of 10 members. From these, GPRASP1-5 have been proposed as one sub-family and GPRASP6-10 as a second sub-family, this distinction is based solely on protein structure analysis (Abu-Helo and Simonin, 2010). GPRASPs have been shown to play a role in endolysosomal degradation of GPCRs, however, this is a still relatively unexplored family of genes. Some landmark work includes the seminal study showing GPRASP1 interacts and targets dopamine receptor toward lysosomal degradation (Whistler et al., 2002); or that via adaptor proteins such as Beclin 2, GPRASP1 is required for ligand-induced degradation of cannabinoid 1 receptor (He et al., 2013).
From structural analysis, sub-family 1 members share significant sequence homology, but it has been hypothesized that discrete GPRASPs may regulate specific receptors (Bockaert et al., 2010), or that GPCR sorting may be performed in a cell-type specific manner (Ritter and Hall, 2009). However, most prior studies looking at potential interactions have not performed functional assays. Here we are now providing novel data showing that GPRASP2 co-localizes with markers of endocytic pathway and that upon DHPG stimulation, GPRASP2 shows significant increase in co-localization with Lamp1 (Supplementary Figure 6). This is conducive with the known role of GPRASPs in targeting GPCRs for lysosomal degradation.
- ii) In terms of their expression, we analyzed the Gtex database to produce an overview of the expression patterns of GPRASPs in the human brain (Supplementary Table 1). We found that GPRASP1-3 are preferentially expressed in the brain, while GPRASP4-10 are mostly expressed in other tissues. Interestingly, GPRASP1 and GPRASP2 seem to show local expression differences in the human brain (data not shown). In rodents, *Gprasp1* was shown to play a role in striatal behaviors while *Gprasp2* is only transiently enriched in this brain region (Supplementary Figure 2j and 2k). Finally, we amplified *Grasp1-10* from the mouse brain and found no difference in their expression levels in the KO brain. We failed to amplify *Gprasp5*, which according to the Gtex database presents very low transcript levels. This data suggest that the *Gprasp* family member regulation is independent from *Gprasp2* and that there is no compensatory upregulation of other *Gprasps* in *Gprasp2* KO mice.

2. It seems arbitrary to hypothesize that GPRASP2 regulates mGluR5. How does GPRASP2 regulate mGluR5? Maybe by direct interaction? If so, Co-IP should be performed. Is the association specific to mGluR5 compared to other mGluRs and other GPCR?

- i) Our original work was guided by data in the literature suggesting GPRASP2 interacts with mGluR5 and the disease-associated data from *GPRASP2* mutations in humans. This interaction was confirmed by our studies showing GPRASP2 manipulations leads to altered mGluR5 levels and functional analysis using DHPG-mediated LTD (Figure 6). To support our work, we followed the

Reviewer's suggestion for further experimentation and we are now providing evidence showing that GPRASP2 and mGluR5 co-immunoprecipitate (Supplementary Figure 6a). We also show that GPRASP2 localization with endocytic markers is influenced by DHPG stimulation (Supplementary Figure 6b). Together, these results considerably strengthen the paper and our conclusions, and we thank the Reviewer for his/her suggestion.

- ii) Regarding the second part of the question, on specificity of GPRASP2-mGluR5 regulation, we made changes to the manuscript to clarify that we are not making a claim regarding specificity. Previous work has shown that GPRASPs may indeed modulate several targets, but without in-depth analysis such as the one we perform here for GPRASP2-mGluR5, it is unknown if those other putative interactions are meaningful or not. As such, the possibility for functional alterations in other GPCRs, potentially in other brain regions, remains unknown and goes beyond the scope of our work, since we cannot prove that GPRASP2 does not interact with all other GPCRs. As such, performing pull-down experiments and showing negative data (e.g. for example that GPRASP2 does not co-IP another type of receptors) could provide a potentially misleading statement. To highlight our point, we have reinforced our previous statements in the introduction (P2L60-L62) and discussion (P8L259-L262) as not to reject the potential regulation of other GPCRs by GPRASP2. In particular, we highlight that at the very least, indirect regulation is to be expected to occur via the promiscuous interaction of mGluR5 with other GPCRs, such as A2A-mGluR5 heterodimers (Tebano et al., 2005). We hope the Reviewer finds our argument balanced and agrees that we are not making a statement regarding specificity, but rather that the preponderance of evidence presented supports our claim that mGluR5 plays an important role in GPRASP2-mediated dysfunction.

3. Fig 1D, which brain region was used? Multiple brain regions including hippocampal and cortical tissue should be used to compare the expression pattern.

In Figure 1D whole brain lysate was used. We have now added a comparison between protein expression in the hypothalamus, hippocampus, cerebellum and cortex (Supplementary Figure 2I). This data supports the previous results from our situ mRNA hybridization showing highest expression in the hippocampus and hypothalamus regions.

4. Fig 1I, changes at the distal region suggests that the alteration is limited to apical dendrite; so that a comparison of the basal vs. apical dendrites will be helpful. Is the change specific for hippocampal CA1 neurons? How about cortical neurons?

- i) We have added the comparison between apical and basal dendrites in the hippocampus. While CA1 pyramidal neurons basal dendrites follows a similar trend, the difference in complexity is not statistically significant. Conversely, the complexity changes is amplified if apical dendrites are analyzed separately (as can be seen in Supplementary Figure 3). We thank the Reviewer for inquiring on this more refined analysis, since it helps highlight the specific alterations found in this KO line.
- ii) Regarding the second part, we opted to analyze hypothalamic neurons instead of cortex, since this region would better match some of our other behavioral findings (e.g. increased body weight) and because of higher GPRASP2 expression in the hypothalamus. Surprisingly, we saw no change in neuronal complexity or spine density in this region. This may be attributed to the already very low complexity of these neurons, heterogeneity in cell types in this region, or to regional redundancy by GPRASP1 or GPRASP3.

5. Fig 2, the mechanism underlying the reduction in mEPSC amplitude was not studied. At least the expression of AMPAR should be explored. Also, the study showed that the dendritic arborization was reduced, and therefore the total synapse number in a neuron is expected to be lower. However, mEPSC frequency was not changed. Why?

We added biochemistry data that indeed supports a change in AMPA receptor levels as the most likely mechanism for the observed reduction in mEPSC amplitude (Figure 2j).

Regarding the apparent inconsistency in dendritic arborization/spine density with the no changes in mEPSC frequency, we would point out that this data is reconciled due to the changes in the KO being mostly in distal dendritic regions (Supplementary Figure 3). As with any technique, whole cell patch-clamp suffers from a few limitations, not the least of which resides in poor spatial clamp outside of the cell soma (Spruston et al., 1993; Williams and Mitchell, 2008). This effect causes electrotonic filtering of events occurring distally from the recording site. As such, the reduced complexity and spine density being more evident in distal sites and our ability to register differences in between WT and KO at those distal locations being muffled under this recording configuration, is

the potential culprit for this lack of convergence. Of course, another possibility could be the fact that dendritic spines account only for a subset (albeit a significant) of all glutamatergic synapses a discrete CA1 neuron receives. Nevertheless, considering the specific morphological alterations in the KOs and the technical characteristics of whole-cell patch clamp, we suggest that this provides the most parsimonious explanation for our data.

6. Fig 3, the density of mature synapse was reduced while that of the immature synapse showed no change. However, the overall synapse density showed no change. What causes the inconsistency?

We thank the Reviewer for pointing out this apparent inconsistency. We re-analyzed our data and found that the separation of the whole population of spines into mature and immature creates two independent populations (i.e. each dendritic shaft does not have a set ratio of mature to immature spines). As such, there is no "statistical carry-over" between analysis and the fact that there are change in one population is occluded by lack of change in another population (which obscures statistical significance when combined).

If any conclusion should be drawn from this observation is that indeed the separation of mature/immature spines must be done as a default analysis. Regardless, we were inspired by the Reviewers previous line of questioning regarding basal (proximal) and apical (distal) changes and we also quantified spine density in basal secondary dendrites. We found a similar trend, but no significant differences in this population. However, when basal and apical data were combined to increase our statistical power, this provided a result of total spine reduction ($p < 0.05$) and a reduction of mature spines as ($p < 0.01$) and no change in immature spines ($p > 0.05$). As above, the conclusion is that most significant morphological differences seem to occur more strongly in distal regions.

7. Fig 4, there is no confirmation on the shRNA knockdown effect on GPRASP2 in neurons. Similarly, it is important to confirm that the GPRASP expression is recovered to the basal level (i.e., rescued). Because the rescue expression might actually lead to GPRASP2 overexpression.

We now provided information regarding knockdown efficiency as well as protein levels for the rescue experiment (Supplementary Figure 5a-c). In the rescue experiment (Supplemental Figure 5) indeed we can observe a slight increase in GPRASP2 area, but not in puncta intensity or puncta number in the rescue conditions. This could potentially explain why there is a small overshoot in some spine morphology parameters in the rescue experiment in Figure 4f-g. However, we would stress that the rescue experiment was performed to confirm the specificity of the shRNA effects following GPRASP2 depletion. Moreover, following indications from Reviewer #2, we also added a second shRNA to solidify our data, which again argues that Gprasp2 knockdown produces specific alterations to neuronal morphology and spine density.

8. What is the possible mechanism for the alterations in dendrite growth and spine formation? Via mGluR5 signaling?

Following lines of questioning from all three Reviewers to provide additional mechanistic insight into GPRASP2 alterations, we decided to test if indeed the dendritic growth and spine density occur due to excessive mGluR5 signaling. The potential for mGluR5 involvement in the above processes is supported by evidence in the literature linking mGluR5 to spine maturation (Oh et al., 2013) and also its role in neurite growth (Mion et al., 2001). To assess if indeed excessive mGluR5 plays a role in neuronal morphological parameters, following knockdown of GPRASP2, we performed shRNA KD while incubating with MPEP, a selective mGluR5 antagonist.

This new data is presented as Supplementary Figure 7 and indicates that MPEP treatment rescues neuronal complexity and spine density changes. These results support the role of enhanced mGluR5 signaling derived from loss of GPRASP2 as a key aspect in GPRASP2 mediated-dysfunction.

9. Fig 6 AB, the effect on mGluR1 was not specifically studied (only studied in combination with mGluR5). What is the underlying mechanisms? Does the change in mGluR cluster result from alterations in total mGluR protein amount, trafficking or synapse maturation? In addition, a non-Group I mGluR member should be used as a negative control.

- i) In Figure 6a-f to detect total vs surface mGluR5 we resorted to a combination of antibodies where one of them also detects mGluR1 in order to detect total mGluR5 levels. We opted to proceed with this experiment since it has been describe that mGluR1 is virtually undetectable (via Northern Blot or Western) in the developing rodent hippocampus (Ryo et al., 1993; van den Pol et al., 1998). Since our experiments are carried out in cultured rat neurons isolated at E18 we do not anticipate that the detection of mGluR1 significantly contributes to overall signal. Regardless, we felt we should indicate the nature of both antibodies as being different in the data panels, even if we are mostly detecting mGluR5.
- ii) In essence, our results speak to significant alterations in the surface levels of mGluR5 together with some alteration to overall mGluR5 levels. This is accordance with the bulk of our data showing a mechanism by which GPRASP2 is important for trafficking during desensitization of mGluR5. In the new experiments we include, we show there is increased co-localization of GPRASP2 with Lamp1 upon DHPG stimulation (Supplementary Figure 6b), which indicates that the canonical GPRASP

mechanism is in play and that receptor desensitization via internalization and degradation is compromised in the absence of GPRASP2, which in turn leads to receptor accumulation at the surface (Figure 6). Conversely, our overexpression data also supports this conclusion since enhanced presence of GPRASP2 shifts the system towards reduced surface levels of mGluR5 even before DHPG application. In sum, the block (with knockdown) or occlusion (with overexpression) of the effect of DHPG upon surface levels of mGluR5 clearly suggests GPRASP2 interferes with both desensitization and trafficking of these receptors.

- iii) The Reviewers is also questioning if the synapse maturation could be playing a role in the enhanced presence of mGluR5, and if it could be a cause, instead of a consequence of the alterations to receptor levels. We would argue that, even though mGluR levels and synapse maturation are inextricably linked, altered synaptic maturation should not perturb mGluR desensitization upon agonist binding. Since there are changes to internalization both with overexpression and knockdown of GPRASP2, the most parsimonious explanation is that indeed GPRASP2 is interfering with receptor trafficking. Moreover, new data (Supplementary Figure 7) shows that MPEP antagonism of mGluR activity during GPRASP2 knockdown blocks changes to spine maturation. This data argues that mGluR signaling is upstream of the alterations in spine morphology.
- iv) Unfortunately we could not perform the experiments suggested and test mGluR Type II and Type III since these receptors are present mostly in the presynaptic compartment of hippocampal synapses (Shigemoto et al., 1997). This would of course preclude any comparison with our data and would not function as intended by the Reviewer. Moreover, we would argue that even if such an experiment were feasible, it would not be a negative control, but rather negative data showing lack of association between GPRASP2 and some other receptor. This again would narrow down the list of targets GPRASP2 does not interact with; but would be an immaterial piece of evidence for the present work. Finally, the procurement and validation of tools necessary for such an experiment is rarely available for most GPCRs since we would need not one, but two exceedingly good antibodies, both working for immunocytochemistry where one is specific for extracellular domain and another for an intracellular epitope.

10 Fig 6G, there is basically no difference except for the small portion after 60min? Is that due to worsened cell conditions? How about LTP and LFS-induced LTD given the changes in hippocampal synaptic change and impaired memory?

- i) The kinetics and magnitude of our results fall within what is seen in the literature for DHPG mediated LTD (Fitzjohn et al., 1999; Xiao et al., 2001) and is in line what is expected for animal models of perturbed mGluR signaling (Auerbach et al., 2011; Klein et al., 2015). In Figure 6G, our analysis is performed across the entire dataset and our RM two-way ANOVA reports statistical differences for the entire trace length (0-80') with a $p=0.019$. We agree that the physiological magnitude of the difference may not appear extremely expressive, but again is in line with what has been described for other animal models showing dysfunction in DHPG-mediated LTD.
- ii) Unfortunately, we did not perform the experiments suggested by the Reviewer since they go beyond the scope of the present study. We feel an in-depth look at plasticity changes is indeed very interesting, but deserving of its own separate analysis.

11. Fig 8, the KO mice showed minor abnormalities in social interaction (Fig 8B). More significant change was detected in social novelty (with stranger 2), but impairments in memory may, at least partially, contribute to the change.

We agree with the Reviewers assessment and modified P8L235-238, to state "Gprasp2-/- mice spent less time overall engaging in social interaction and also displayed a lower index of preference for the novel stimulus (Fig. 8 d-e), however, the lack of preference for the 'S2' partner may result from both social recognition deficits compounded with the memory impairments displayed by Gprasp2-/- mice."

Reviewer #2 (Remarks to the Author):

We thank the Reviewer for his/her expert analysis of this work and for encouraging comments regarding novelty and interest for our study. We are also grateful for suggestions which have guided new experiments added to the manuscript.

In this manuscript, Peca et al. explore the function of Gprasp2, a little-studied GPCR sorting protein linked to ASD and ID. Overall, this study represents a broad and thorough characterization of a constitutive knock-out allele. Given the association of this molecule with human disease, as well as its potential ability to bind a wide range of GPCRs throughout the body and brain, I think this work would be of broad interest to the neuroscience field. There is a small literature regarding the function of this molecule and this work would make a significant contribution to that knowledge. The experiments represent standard approaches in the field and appear well executed, soundly interpreted and nicely illustrated. Overall, the statistical tests are typical for these assays.

Currently, there are a few issues that should be addressed to strengthen and round out this story:

Major Points:

1. the work in dissociated culture does not dramatically enhance the previous findings and raises far more issues than it solves. First, if the authors were interested in exploring the cell autonomous nature of the phenotype, why didn't they just use their conditional allele and the elegant AAV9 mosaic infection approach? This would allow us to continue to compare apples and apples, as opposed to now having to compare in vivo manipulations in an intact mouse brain with dissociated cultures (from another species) that are undergoing a new wave of synaptogenesis within the dish. Using this AAV9 approach, they could have also injected at different times to get at questions of compensation as well as distinct temporal roles for this protein (development vs mature function). As it stands, it creates the need to reconcile the specific effects on mature spines of the KO from the broad spine changing effects of the shRNA - what do these differences mean?, do they reflect developmental vs. maintenance phenotypes? Furthermore, this dendrite and spine density phenotype is always a potential problem when using shRNAs in hippocampal cultures (Alvarez et al. JNeuro. 2006), requiring very strong controls. Along these lines, the authors do use a scrambled shRNA as well as a WT replacement. However, the replacement results are difficult to interpret in light of the spine density GOF phenotype (Fig5) because you may just additively be getting rid of off-target effects. In fact, it does seem like the KD+rescue has lower levels of spines (Fig4E) than the O/E alone (Fig5E). alternatives to prove lack of off-target effects would be another KD showing the same phenotype or using the KD construct employed in Fig4 on the KO mouse background (and seeing no dendrite length/spine density phenotype). Overall, the strongest point to be taken from the in vitro work is that Gprasp2 can modulate surface levels of mGluR5. Even here though, it does seem that some of this is hard to interpret given that some of the manipulations also cause global changes in mGluR5 protein levels (Fig 6E,F). These concerns about cell autonomy and compensation are valid but could be addressed more strongly.

The reviewer argues two points, on one hand that the *in vivo* data and *in vitro* data do not perfectly match and raises a second major point in that the shRNAs are unreliable as they require very strong controls.

Regarding the first point, one of the key points in our rationale to setting up the *in vitro* culture system is that it segues into the mGluR5 analysis in cultured neurons. These allowed us to perform functional assays and draw convincing results regarding the role of GPRASP2 on surface mGluR5 levels upon DHPG-mediated stimulation. Also, the culture system allowed us to perform overexpression studies to determine that changing levels of GPRASP2 bidirectionally alters spine density and dendritic morphology. As it stands, the magnitude of the changes are indeed dissimilar (e.g. Spine density: -45% in KD; -25% in KO), but their direction is the same. As such we argue that the culture system and the *in vivo* model do not necessarily require reconciliation. As it is widely abundant in the literature, culture systems tend to amplify (or the *in vivo* to adapt) to whatever changes are imprinted on neurons. For example, almost all major postsynaptic density proteins involved in autism have been shown to produce dramatic and reproducible effects in spines and synapses in dissociated neuron cultures. Manipulation of neuroligins, in particular, were one of the first such examples to show dramatic changes in culture (Chih et al., 2005). However, the KO mice for NLs do not display similar magnitude of alterations. This was famously discussed by Thomas Sudhof and Nils Brose in Varoqueaux et al., 2006. The arguments made by the authors revolve around indirect and activity-dependent homeostatic effects that are amplified in culture system due very low changes in activity levels (Turrigiano and Nelson, 2004). They conclude that "knock-down of NLs in cultured neurons affects mainly labile and inactive synapses, which might even be unique to cultured neurons". To this we would add the growing evidenced that microglia plays a critical role in synaptic physiology and stability (Paolicelli et al., 2011). Since this microglia are actively removed in Banker cultures (Kaech and Banker, 2006),

perhaps these cells also play a role in the difference expressed between systems. While we agree with the Reviewer that the *in vitro* and *in vivo* data are not directly comparable, precisely the fact that a similar dysfunction arises when analyzing two different systems across two different experimental procedures, gives us even stronger confidence that effects produced by KO or KD or *Gprasp2* are indeed physiologically very relevant.

Regarding the Reviewer's second point, we agree that the shRNAs require careful validation. The suggested KD experiment in KO background mouse neurons, while an elegant study was excluded because we are using rat neuron cultures and rat shRNA. Since the shRNAs are expected to display potential species-specific off-target effects (Jackson and Linsley, 2010), this would not allow us to draw a strong conclusion in mouse KO cultures, even in the absence of cellular phenotype.

As such, we followed the Reviewer's second suggestion and add another shRNA, targeting a different region of the rat *Gprasp2* transcript. As can be observed in Supplementary Figure 6 we obtained a largely similar results as to the one originally shown. We believe adding this data strengthened our results considerably. As they again indicate there is a clear dose-response dependency to the effects of removing or adding more GPRASP2.

The Reviewer also mentions the changes in global mGluR5 levels as confounding the interpretation of the results. We argue that this is actually in line with the proposed canonical role for GPRASP2 in the trafficking of GPCRs for lysosomal degradation. We have added new data showing that indeed upon DHPG stimulation, GPRASP2 strongly co-localizes with Lamp1, suggesting that the final output of GPRASP2 interaction following receptor desensitization is geared towards receptor degradation.

2. while mentioned in the text, one major issue here is the implication of mGluR specificity for a protein whose specificity towards its GPCR binding partners is not clear. While I believe the authors that this mutation can alter mGluR surface levels, the paper is written in a way to suggest that this is a likely explanation for much of what follows, both at the morphological and behavioral level. Because this link with mGluR signaling enhances the profile and translation relevance of the work, I would suggest the authors try to draw a more concrete link. To do this, what about taking the mutant and using either systemic or targeted negative modulators of mGluR5 signaling to attempt to ameliorate either a morphological (probably easier) or behavioral phenotype. If you have a strong hypothesis that the deficit is developmental in origin, perhaps just using this modulation during neuronal culturing and looking at morphology?

We thank the Reviewer for suggesting this experiment. We performed MPEP incubation in the neuronal culture system and showed that it rescues the spine and neuronal morphology alterations (Supplementary Figure 7). This is in line with a role for mGluR5 in spine maturation (Oh et al., 2013) and in neurite outgrowth (Mion et al., 2001). This experiment supports our hypothesis that alterations in mGluR5 mediate several of the observed cellular morphology in *Gprasp2* loss of function.

Minor Points:

1. Fig 1: I don't see much evidence for specificity of expression, either at P15 or at P5 in the striatum. This molecule looks widely expressed. Regional qPCR would be more convincing if one really wanted to show this.

We slightly modified the text to more clearly state that the expression in the striatum is enriched (not specific to this brain region). The same was done regarding the enrichment in hippocampus and hypothalamus which can be seen in adult mouse brain. We respectfully guide the Reviewer to Supplementary Figure 2 to assess the regional levels of *Gprasp2* in the *in situ* mRNA hybridization experiments.

2. Fig2E: bad trace example, shows the reduced amplitude but also has slower kinetics.

We have now placed the two traces, normalized along the y-axis, in order to allow a better assessment of the kinetics.

3. Fig3A: missing 'A' label.

We thank the Reviewer for bringing this to our attention. We have corrected it in the new version.

4. There are numerous grammatical errors throughout the text that should be fixed.

We performed a more careful proofreading of the revised manuscript and remain available to subscribe to the Springer-Nature Author Services if the Editor entertains the possibility of accepting our manuscript for publication.

Reviewer #3 (Remarks to the Author):

We thank the Reviewer for his/her careful reading of the manuscript, suggestions for experiments and for endorsing the publication or our work.

This is a timely and important study examining a mutant mouse model with a deletion of the GPCR interacting protein Gprasp2. Human mutations in this gene have been linked to autism and intellectual disability. In this study, the authors examine the neurological phenotypes associated with loss of Gprasp2 especially with respect to modulation of group 1 mGluRs, which are implicated in autism and ID.

This study is impressive, using multiple methodologies to examine biochemical, structural, electrophysiological and behavioural phenotypes in the Gprasp2 mutant mouse. The overall conclusions of the authors are that reduction of Gprasp2 leads to significant disruptions in synaptic structure and function in hippocampal neurons, and an exaggeration of mGluR-mediated plasticity that may be due to a change in the trafficking of mGlu5 receptors. This in turn leads to deficits in learning and social behaviour reminiscent of other autism/ID mouse models.

It is clear that the authors were rigorous in the experimental design and analysis for each part of this study, and this reviewer has no reservations for recommending it for publication. However, a few additions might strengthen the manuscript:

1.) The overexpression of mGlu5 at the surface suggests that the intracellular signalling downstream of this receptor should be enhanced. Have the authors tested whether this is the case? Do they believe that other functions of group 1 mGluRs might be exaggerated?

See below.

2.) Have the authors thought of testing whether negative allosteric modulators such as MTEP or CTEP could reverse phenotypes in the Gprasp2 mutant?

The Reviewer wants to know if 1) other function of mGluR5 are enhanced and 2) if mGluR5 antagonists may revert any of the observed phenotypes. Taking the Reviewers questions into consideration, we proposed to first test *in vitro* if enhanced mGluR5 could be playing a role in dendritic elongation and spine density since these receptors are closely linked to Shank and other proteins involved in actin polymerization. This would be consistent with reports showing that mGluR5 isoform bidirectionally control neurite outgrowth (Mion et al., 2001) and that mGluR5 KO display enhanced dendritic spine density (Chen et al., 2012). With those results in mind we hypothesize that during development, GPRASP2 modulation of surface receptor levels may help regulate these processes by controlling surface levels of mGluR5. To assess if indeed GPRASP2 knockdown-mediated reduction in dendritic arborization and spine density was stemming from increased intracellular mGluR5 activity, we incubated neuronal cultures for 5 days with MPEP after transfection with shRNA. We observed that blocking mGluR5 activity prevented reduction to neuronal arborization and decreased spine density (Supplementary Figure 7).

Future work will aim at dissecting potential changes to signaling proteins as well as alterations to protein synthesis as responsible for perturbing synaptic and cellular changes. Regarding this, we have already observe that PSD-95 is downregulated in KO mice, while CamK2a shows a trend towards increase.

3.) Other GPCR interacting proteins have been implicated in autism and ID (ie, Homer1, arrestins). Is there a phenotypic similarity between these other models and the Gprasp2 mutant?

This is a very interesting question! Indeed, one of the similar phenotypes we highlighted in this work is that GPRASP2 KOs display lower percentage of reciprocal social interaction and higher percentage of victories in the tube test. This could suggest that GPRASP2 mice are not only less sociable, they are more aggressive in their interactions. Together with perturbed memory function in those animals, both these phenotypes are also displayed by Homer1 KO mice. Moreover, it has been proposed that Homer1 KO mice display alteration in their circadian behavior (Jaubert et al., 2007) and that Homer1 expression is regulated by light (Park et al., 1997). This is particularly interesting because GPRASP was actually originally discovered via its interaction with Per1 (the circadian clock regulator) (Matsuki et al., 2001). The high expression level of GPRASP2 in the hypothalamus, the similar phenotypes to Homer1 and their shared proximity to mGluR5 are prompting us to investigate circadian dysfunction in GPRASP2 KO mice.

4.) Given the role of mGlu5 in anxiety, it is interesting that there appears to be no anxiety phenotype in the Gprasp2 mutant. Do the authors have any thoughts on why this might be the case?

Indeed, we find no overt changes in anxiety in *Gprasp2* KO mice. This may be due to GPRASP2 providing a finetune control over mGluR5 surface expression, which unlike an agonist or antagonist, does not necessarily produce receptor activation, but rather modulation of internalization upon ligand binding. We could speculate that perhaps the most similar action to the loss of GPRASP2 would be akin to a positive allosteric modulator (PAM), in that it could prolong meaningful physiological signaling. CDPPB, for example, is an mGluR5 PAM and also produces either no change in anxiety or a small anxiolytic effect when given to mice (Menard et al., 2013). Furthermore, there is also potential cell type and circuit-specific defect only where *Gprasp2* overlaps with mGluR5. This combination of match (cells that are expected to express both proteins) and mismatch (cells that only expressed one of the proteins) may give rise to complicated behavioral interactions.

References

- Abu-Helo, A., and Simonin, F. (2010). Identification and biological significance of G protein-coupled receptor associated sorting proteins (GASPs). *Pharmacol. Ther.* 126, 244–50. doi:10.1016/j.pharmthera.2010.03.004.
- Auerbach, B. D., Osterweil, E. K., and Bear, M. F. (2011). Mutations causing syndromic autism define an axis of synaptic pathophysiology. *Nature* 480, 63–68. doi:10.1038/nature10658.
- Bockaert, J., Perroy, J., Bécamel, C., Marin, P., and Fagni, L. (2010). GPCR interacting proteins (GIPs) in the nervous system: Roles in physiology and pathologies. *Annu. Rev. Pharmacol. Toxicol.* 50, 89–109. doi:10.1146/annurev.pharmtox.010909.105705.
- Chen, C. C., Lu, H. C., and Brumberg, J. C. (2012). mGluR5 knockout mice display increased dendritic spine densities. *Neurosci. Lett.* doi:10.1016/j.neulet.2012.07.014.
- Chih, B., Engelman, H., and Scheiffele, P. (2005). Control of excitatory and inhibitory synapse formation by neuroligins. *Science* 307, 1324–8. doi:10.1126/science.1107470.
- Fitzjohn, S. M., Kingston, A. E., Lodge, D., and Collingridge, G. L. (1999). DHPG-induced LTD in area CA1 of juvenile rat hippocampus; characterisation and sensitivity to novel mGlu receptor antagonists. *Neuropharmacology*. doi:10.1016/S0028-3908(99)00123-9.
- He, C., Wei, Y., Sun, K., Li, B., Dong, X., Zou, Z., et al. (2013). Beclin 2 functions in autophagy, degradation of G protein-coupled receptors, and metabolism. *Cell* 154, 1085–1099. doi:10.1016/j.cell.2013.07.035.
- Jackson, A. L., and Linsley, P. S. (2010). Recognizing and avoiding siRNA off-target effects for target identification and therapeutic application. *Nat. Rev. Drug Discov.* doi:10.1038/nrd3010.
- Jaubert, P. J., Golub, M. S., Lo, Y. Y., Germann, S. L., Dehoff, M. H., Worley, P. F., et al. (2007). Complex, multimodal behavioral profile of the Homer1 knockout mouse. *Genes, Brain Behav.* doi:10.1111/j.1601-183X.2006.00240.x.
- Kaech, S., and Banker, G. (2006). Culturing hippocampal neurons. *Nat. Protoc.* 1, 2406–2415. doi:10.1038/nprot.2006.356.
- Klein, M. E., Castillo, P. E., and Jordan, B. A. (2015). Coordination between translation and degradation regulates inducibility of mGluR-LTD. *Cell Rep.* doi:10.1016/j.celrep.2015.02.020.
- Matsuki, T., Kiyama, A., Kawabuchi, M., Okada, M., and Nagai, K. (2001). A novel protein interacts with a clock-related protein, rPer1. *Brain Res.* doi:10.1016/S0006-8993(01)02857-8.
- Menard, C., Tse, Y. C., Cavanagh, C., Chabot, J.-G., Herzog, H., Schwarzer, C., et al. (2013). Knockdown of Prodynorphin Gene Prevents Cognitive Decline, Reduces Anxiety, and Rescues Loss of Group 1 Metabotropic Glutamate Receptor Function in Aging. *J. Neurosci.* doi:10.1523/JNEUROSCI.0290-13.2013.
- Mion, S., Corti, C., Neki, A., Shigemoto, R., Corsi, M., Fumagalli, G., et al. (2001). Bidirectional regulation of neurite elaboration by alternatively spliced metabotropic glutamate receptor 5 (mGluR5) isoforms. *Mol. Cell. Neurosci.* 17, 957–972. doi:10.1006/mcne.2001.0993.
- Oh, W. C., Hill, T. C., and Zito, K. (2013). Synapse-specific and size-dependent mechanisms of spine structural plasticity accompanying synaptic weakening. *Proc. Natl. Acad. Sci.* 110, E305–E312. doi:10.1073/pnas.1214705110.
- Paolicelli, R. C., Bolasco, G., Pagani, F., Maggi, L., Scianni, M., Panzanelli, P., et al. (2011). Synaptic pruning by microglia is necessary for normal brain development. *Science (80-)*. doi:10.1126/science.1202529.
- Park, H. T., Kang, E. K., and Bae, K. W. (1997). Light regulates Homer mRNA expression in the rat suprachiasmatic nucleus. *Mol. Brain Res.* doi:10.1016/S0169-328X(97)00292-1.
- Ritter, S. L., and Hall, R. A. (2009). Fine-tuning of GPCR activity by receptor-interacting proteins. *Nat. Rev. Mol. Cell Biol.* 10, 819–30. doi:10.1038/nrm2803.
- Ryo, Y., Miyawaki, A., Furuichi, T., and Mikoshiba, K. (1993). Expression of the metabotropic glutamate receptor mGluR1 α and the ionotropic glutamate receptor GluR1 in the brain during the postnatal development of normal mouse and in the cerebellum from mutant mice. *J. Neurosci. Res.* doi:10.1002/jnr.490360104.
- Shigemoto, R., Kinoshia, A., Wada, E., Nomura, S., Ohishi, H., Takada Masahiko, et al. (1997). Differential Presynaptic Localization of Metabotropic Glutamate Receptor Subtypes in the Rat Hippocampus. *J. Neuroscience*. doi:9295396.
- Spruston, N., Jaffe, D. B., Williams, S. H., and Johnston, D. (1993). Voltage- and space-clamp errors associated with the measurement of electrotonically remote synaptic events. *J. Neurophysiol.* doi:10.1152/jn.1993.70.2.781.
- Tebano, M. T., Martire, A., Rebola, N., Pepponi, R., Domenici, M. R., Grò, M. C., et al. (2005). Adenosine A2A receptors and metabotropic glutamate 5 receptors are co-localized and functionally interact in the hippocampus: A possible key mechanism in the modulation of N-methyl-D-aspartate effects. *J. Neurochem.* doi:10.1111/j.1471-4159.2005.03455.x.
- Turrigiano, G. G., and Nelson, S. B. (2004). Homeostatic plasticity in the developing nervous system. *Nat. Rev. Neurosci.* 5, 97–107. doi:10.1038/nrn1327.
- van den Pol, A. N., Gao, X. B., Patrylo, P. R., Ghosh, P. K., and Obrietan, K. (1998). Glutamate inhibits GABA excitatory activity in developing neurons. *J. Neurosci.* doi:10.1523/JNEUROSCI.18-24-10749.1998.
- Varoqueaux, F., Aramuni, G., Rawson, R. L., Mohrmann, R., Missler, M., Gottmann, K., et al. (2006). Neuroligins

- Determine Synapse Maturation and Function. *Neuron* 51, 741–754. doi:10.1016/j.neuron.2006.09.003.
- Whistler, J. L., Enquist, J., Marley, A., Fong, J., Gladher, F., Tsuruda, P., et al. (2002). Modulation of postendocytic sorting of G-protein coupled receptors. *Science (80-.)*. 297, 615–620.
- Williams, S. R., and Mitchell, S. J. (2008). Direct measurement of somatic voltage clamp errors in central neurons. *Nat. Neurosci.* 11, 790–798. doi:10.1038/nn.2137.
- Xiao, M. Y., Zhou, Q., and Nicoll, R. A. (2001). Metabotropic glutamate receptor activation causes a rapid redistribution of AMPA receptors. *Neuropharmacology*. doi:10.1016/S0028-3908(01)00134-4.

REVIEWERS' COMMENTS:

Reviewer #1 (Remarks to the Author):

The authors have addressed all of my concerns. The paper is significantly strengthened by new experiments, additional data analysis and text clarification. The current version is likely appropriate for publication.

Reviewer #2 (Remarks to the Author):

The authors have experimentally addressed my two major concerns - the specificity of the RNAi and the mechanistic role mGluR signaling. the second RNAi while achieving less KD of Gprasp2 still produces largely similar morphological phenotypes. furthermore, the MPEP experiment is a solid step in linking mGluR signaling together with the observed neural phenotypes.

on this basis, i recommend publication.

Reviewer #3 (Remarks to the Author):

The authors have done a very thorough job addressing all criticisms, and the study has been greatly strengthened. I have no hesitations recommending this for publication.

REVIEWERS' COMMENTS:

Reviewer #1 (Remarks to the Author):

The authors have addressed all of my concerns. The paper is significantly strengthened by new experiments, additional data analysis and text clarification. The current version is likely appropriate for publication.

We thank the Reviewer for his/her expert line of questioning which helped guide the experiments added to the manuscript.

Reviewer #2 (Remarks to the Author):

The authors have experimentally addressed my two major concerns - the specificity of the RNAi and the mechanistic role mGluR signaling. the second RNAi while achieving less KD of Gprasp2 still produces largely similar morphological phenotypes. furthermore, the MPEP experiment is a solid step in linking mGluR signaling together with the observed neural phenotypes.

on this basis, i recommend publication.

We thank the Reviewer for suggesting critical experiments that help support the conclusions of our work.

Reviewer #3 (Remarks to the Author):

The authors have done a very thorough job addressing all criticisms, and the study has been greatly strengthened. I have no hesitations recommending this for publication.

We thank the Reviewer for his line of questioning. The comments he/she put forth are already leading into interesting avenues.